# Robust Deep Reinforcement Learning through Adversarial Loss

**Tuomas Oikarinen**[*]
UC San Diego CSE

**Wang Zhang**
MIT MechE

**Alexandre Megretski**
MIT EECS

**Luca Daniel**
MIT EECS

**Tsui-Wei Weng**
UC San Diego HDSI

## Abstract

Recent studies have shown that deep reinforcement learning agents are vulnerable to small adversarial perturbations on the agent's inputs, which raises concerns about deploying such agents in the real world. To address this issue, we propose RADIAL-RL, a principled framework to train reinforcement learning agents with improved robustness against $l_p$-norm bounded adversarial attacks. Our framework is compatible with popular deep reinforcement learning algorithms and we demonstrate its performance with deep Q-learning, A3C and PPO. We experiment on three deep RL benchmarks (Atari, MuJoCo and ProcGen) to show the effectiveness of our robust training algorithm. Our RADIAL-RL agents consistently outperform prior methods when tested against attacks of varying strength and are more computationally efficient to train. In addition, we propose a new evaluation method called *Greedy Worst-Case Reward* (GWC) to measure attack agnostic robustness of deep RL agents. We show that GWC can be evaluated efficiently and is a good estimate of the reward under the worst possible sequence of adversarial attacks. All code used for our experiments is available at `https://github.com/tuomaso/radial_rl_v2`.

## 1   Introduction

Deep learning has achieved enormous success on a variety of challenging domains, ranging from computer vision [1], natural language processing [2] to reinforcement learning (RL) [3, 4]. Nevertheless, the existence of adversarial examples [5] indicates that deep neural networks (DNNs) are not as robust and trustworthy as we would expect, as small and often imperceptible perturbations can result in misclassifications of state-of-the-art DNNs. Unfortunately, adversarial attacks have also been shown possible in deep reinforcement learning, where adversarial perturbations in the observation space and/or action space can cause arbitrarily bad performance of deep RL agents [6, 7, 8]. As deep RL agents are deployed in many safety critical applications such as self-driving cars and robotics, it is of crucial importance to develop robust training algorithms (a.k.a. defense algorithms) such that the resulting trained agents are robust against adversarial (and non-adversarial) perturbation.

Many heuristic defenses have been proposed to improve robustness of DNNs against adversarial attacks for image classification tasks, but they often fail against stronger adversarial attack algorithms. For example, [9] showed that 13 such defense methods (recently published at prestigious conferences) can all be broken by more advanced attacks. One emerging alternative to heuristic defenses is

---

[*]correspondence to: toikarinen@ucsd.edu, lweng@ucsd.edu

35th Conference on Neural Information Processing Systems (NeurIPS 2021).

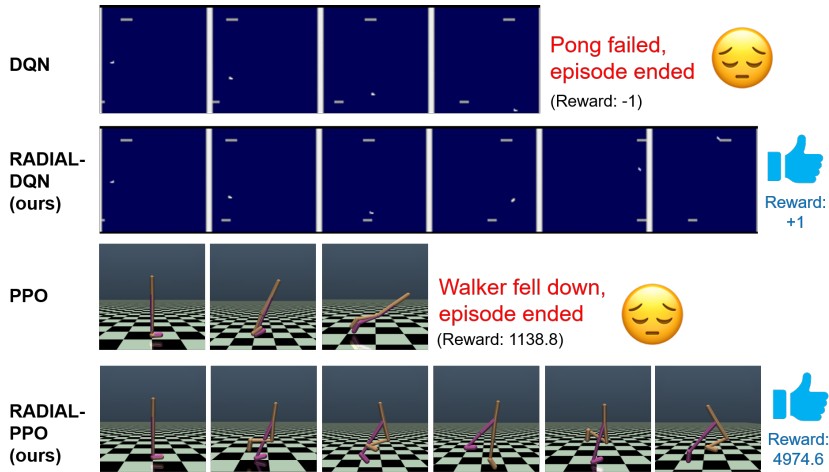

Figure 1: Screenshots of our **RADIAL** framework promoting robustness of deep RL agents while the standard deep RL agents without robust training are vulnerable to adversarial attacks.

defense algorithms [2] based on robustness verification or certification bounds [10, 11, 12, 13]. These algorithms produce robustness certificates such that for any perturbations within the specified $\ell_p$-norm distance $\epsilon$, the trained DNN will produce consistent classification results on given data points. Representative works along this line include [11] and [14], where the learned models can have much higher certified accuracies by including the robustness verification bounds in the loss function with proper training schedule, even if the verifier produces loose robustness certificates [14] for models without robust training (a.k.a. nominal models). Here, the certified accuracy is calculated as the percentage of the test images that are guaranteed to be classified correctly under a given perturbation magnitude $\epsilon$.

However, most of the defense algorithms are developed for classification tasks. Few defense algorithms have been designed for deep RL agents perhaps due to the additional challenges in RL that are not present in classification tasks, including credit assignment and lack of a stationary training set. To bridge this gap, in this paper we present the **RADIAL**(**R**obust **AD**versar**IA**l **L**oss)-RL framework to train robust deep RL agents. We show that **RADIAL** can improve the robustness of deep RL agents by using carefully designed adversarial loss functions based on robustness verification bounds. Our contributions are listed below:

- We propose a novel robust deep RL framework, **RADIAL**-RL, which can be applied to different types of deep RL algorithms. We demonstrate **RADIAL** on three popular RL algorithms, DQN [3], A3C [15] and PPO [16].
- We demonstrate the superior performance of **RADIAL** agents on both Atari games and continuous control tasks in MuJoCo: our agents are $2 - 10\times$ more computationally efficient to train than [17] and can resist up to $5\times$ stronger adversarial perturbations better than existing works [18, 17].
- We also evaluate the effects of robust training on the ability of agents to generalize to new levels using the ProcGen benchmark, and show that our training also increases robustness on unseen levels, reaching high rewards even against $\epsilon = 5/255$ PGD-attacks.
- We propose a new evaluation method, *Greedy Worst-Case Reward* (GWC), for efficiently (in linear time) evaluating RL agent performance under attack of strongest adversaries (i.e. worst-case perturbation) on discrete action environments.

---

[2]We don't use the naming convention in this field to call this type of defense as *certified defense* because we think it is misleading as such defense methodology cannot provide any certificates on unseen data.

## 2 Related work and background

### 2.1 Adversarial attacks in Deep RL

The topic of adversarial examples in supervised learning tasks has been extensively studied, especially for DNN classifiers [5, 19]. More recently, [6, 7, 8] showed that deep RL agents are also vulnerable to adversarial perturbations, including adversarial perturbations on agents' observations and actions [6, 7, 8], mis-specification on the environment [20], adversarial disruptions on the agents [21] and other adversarial agents [22]. For a review of different attack and defense settings in RL see Ilahi *et al.* [23].

In this paper, we focus on $\ell_p$-norm adversarial perturbations on agents' observations since this threat model is adopted by many of the existing works investigating adversarial robustness of deep RL [6, 7, 8, 24, 25, 26, 27, 18, 17]. However, our framework is not limited to $\ell_p$-norm perturbation and in fact can be easily extended to semantic perturbations (e.g. rotations, color/brightness changes, etc) for vision-based deep RL agents (e.g. atari games) by leveraging the techniques proposed in [28].

### 2.2 Formal verification and robust training for Deep RL

Robustness certification methods for DNN classifiers [12] have been applied in the deep RL setting: for example, [27] propose a policy of choosing the action with highest certified lower bound Q-value during execution, and [24] derived tighter robustness certificate for neural network policies under persistent adversarial perturbations in the system. These works study the robustness with fixed neural networks, while our work is focused on training neural networks that produce more robust RL policies.

The idea of adversarial training has been applied to deep RL to defend against adversarial attacks [25, 26]; however, these approaches often have much higher computational cost than standard training. While previous work such as [21] have also trained robust agents under different threat models from ours, we will not be comparing against them as they have different goals and evaluation methods.

The most relevant literature to our work are two robust training methods for deep Q-learning agents [18, 17]. RS-DQN [18] decouples the DQN agent into a policy and student networks, which enables leveraging additional constraints on the student DQN without strongly affecting learning of the correct Q-function, whereas SA-DQN [17] adds a hinge loss regularizer to encourage the DQN agents to follow their original actions when there are perturbations in the observation space. [17] also propose SA-PPO and SA-DDPG for training robust RL-agents in continuous control. In contrast, in **RADIAL**, we leverage the in-expensive robustness verification bounds to carefully design regularizers discouraging potentially overlapping actions of deep RL agents. As demonstrated in the Sec 4, our **RADIAL** agents outperform [18, 17] under various strength of adversarial attacks, while being 2-10× more computationally efficient to train than [17]. Moreover, we introduce a novel metric to evaluate agent performance against worst possible adversary that is not investigated in [18, 17].

### 2.3 Basics of Deep Reinforcement Learning

Markov Decision Process with parameters $(\mathcal{S}, \mathcal{A}, \mathcal{P}, \mathcal{R}, \gamma, s_0)$ is used to characterize the environments in this paper, where $\mathcal{S}$ is a set of states, $\mathcal{A}$ is a set of the available actions, $\mathcal{P} : \mathcal{S} \times \mathcal{A} \times \mathcal{S} \to \mathbb{R}$ defines the transition probabilities, and $\mathcal{R} : \mathcal{S} \times \mathcal{A} \to \mathbb{R}$ is the scalar reward function, $s_0$ is the initial state distribution and $\gamma$ is the discount factor. RL algorithms aim at learning a possibly stochastic policy $\pi : \mathcal{S} \times \mathcal{A} \to \mathbb{R}$ describing the probability of taking an action $a$ given state $s$. The goal of a policy is to maximize the cumulative time discounted reward of an episode $\sum_t \gamma^t r_t$, where $t$ is the timestep and $r_t$, $a_t$ and $s_t$ are reward, action and state at timestep $t$.

**Deep Q-networks (DQN) [3].** An action-value function $Q(s, a)$ describes the expected cumulative rewards given current state $s$ and action $a$. In Q-learning, a policy $\pi$ is constructed by taking the action with highest Q-value. The optimal $Q$ function, denoted as $Q^*(s, a)$, satisfies the Bellman Optimality Equations $Q^*(s, a) = r + \gamma \mathbb{E}_{(s'|s,a)} [\max_{a'} Q^*(s', a')]$, where $s'$ is the next state and $r$ is reward. The essence of DQN is to use neural networks to approximate the $Q^*(s, a)$ and the networks can be trained by minimizing the loss $\mathcal{L}(\theta) = \mathbb{E}_{(s,a,s',r)} [(r + \gamma \max_{a'} Q(s', a'; \theta) - Q(s, a; \theta))^2]$. The two baseline works on robust training for RL agents [18, 17] use more advanced versions than vanilla

DQN including Dueling-DQN [29] and Double-DQN [30]. Double-DQN uses two Q-networks to evaluate target value $Q_{\text{target}}$ and the one being trained $Q_{\text{actor}}$, with $\theta_{\text{actor}}$ being optimized by minimizing the loss $\mathcal{L}(\theta_{\text{actor}})$:

$$\mathcal{L}(\theta_{\text{actor}}) = \mathbb{E}_{(s,a,s',r)} \left[ (r + \gamma \max_{a'} Q_{\text{target}}(s', a'; \theta_{\text{target}}) - Q_{\text{actor}}(s, a; \theta_{\text{actor}}))^2 \right]. \tag{1}$$

Dueling-DQN improves DQN by splitting Q-values into the value of the state $V_Q(s)$ and advantage $A_Q(s, a)$ calculated by different output layers such that $Q(s, a) = V_Q(s) + A_Q(s, a)$.

**Asynchronous Advantage Actor Critic (A3C) [15].** A3C uses neural networks to learn a policy function $\pi(a|s; \theta)$ and a state-value function $V(s; \theta_v)$. Here the policy network $\pi(a|s; \theta)$ determines which action to take, and value function evaluates how good each state is. To update the network parameters $(\theta, \theta_v)$, an estimate of the advantage function, $A_t$, is defined as $A_t(s_t, a_t; \theta, \theta_v) = \sum_{i=0}^{k-1} \gamma^i r_{t+i} + \gamma^k V(s_{t+k}; \theta_v) - V(s_t; \theta_v)$ with hyperparameter $k$. The network parameters $(\theta, \theta_v)$ are learned by minimizing the following loss function:

$$\mathcal{L}(\theta, \theta_v) = \mathbb{E}_{(s_t, a_t, r_t)} \left[ A_t^2 - A_t \log \pi(a_t) - \beta \mathcal{H}(\pi) \right], \tag{2}$$

where the first term optimizes the value function, second optimizes policy function and last term encourages exploration by rewarding high entropy $\mathcal{H}$ of the policy with scaling parameter $\beta$.

**Proximal Policy Optimization (PPO) [16].** PPO is a critic based policy-gradient method similar to A3C, that works for off-policy updates unlike A3C.

PPO uses the clipping function on the ratio of action probabilities to constrain the difference between new and old policy. The resulting objective function is

$$\mathcal{L}(\theta) = \mathbb{E}_{(s_t, a_t, r_t)} \left[ -\min(\frac{\pi(a_t|s_t; \theta)}{\pi(a_t|s_t; \theta_{\text{old}})} A_t, \text{clip}(\frac{\pi(a_t|s_t; \theta)}{\pi(a_t|s_t; \theta_{\text{old}})}, 1 - \eta, 1 + \eta) A_t) \right], \tag{3}$$

where $\eta$ is a hyper-parameter and $A_t$ is an estimate of the advantage function at timestep $t$. In addition, PPO usually includes a term minimizes loss of the value function and rewarding entropy of the policy similar to A3C.

## 3 A Robust Deep RL framework with adversarial loss

In this section, we propose **RADIAL** (**R**obust **AD**versar**IA**l **L**oss)-RL, a principled framework for training deep RL agents robust against adversarial attacks. **RADIAL** designs adversarial loss functions by leveraging existing neural network robustness formal verification bounds. We first introduce the key idea of **RADIAL** and then elucidate a few ways to formulate adversarial loss for the three classical deep reinforcement learning algorithms, DQN, A3C and PPO in Sections 3.1, 3.2 and 3.3. In Section 3.4, we propose a novel evaluation metric, *Greedy Worst-Case Reward* (GWC), to efficiently assess agent's robustness against input perturbations.

**Main idea.** The training loss of the **RADIAL** framework, $\mathcal{L}_{\text{RADIAL}}$, consists of two terms:

$$\mathcal{L}_{\text{RADIAL}} = \kappa \mathcal{L}_{\text{nom}} + (1 - \kappa) \mathcal{L}_{\text{adv}}, \tag{4}$$

where $\mathcal{L}_{\text{nom}}$ denotes the nominal loss function such as Eqs. (1)-(3) for nominal (standard) deep RL agents, and $\mathcal{L}_{\text{adv}}$ denotes the adversarial loss which we will design carefully to account for adversarial perturbations. $\kappa$ is a hyperparameter controlling the trade-off between standard performance and robust performance with value between 0 and 1, and note that standard RL training algorithms have $\kappa = 1$ throughout the full training process. Here we propose two principled approaches to construct $\mathcal{L}_{\text{adv}}$ both via the neural network robustness certification bounds [14, 12, 13, 11, 31, 32, 33, 34]:

**#1.** Construct an *strict* upper bound of the perturbed standard loss (Eq (9), (10), (7));

**#2.** Design a regularizer to minimize overlap between output bounds of actions with large difference in outcome (Eq (5), (6)).

The Approach **#1** is well-motivated as minimizing a strict upper bound of the perturbed standard loss usually also decreases the true perturbed standard loss, which indicates the policy should perform

well under adversarial perturbations. Alternatively, Approach **#2** is motivated by the idea that we want to avoid choosing a significantly worse action because of a small input perturbation.

The foundation of both Approach **#1** and **#2** lies in the robustness formal verification tools to derive output bounds of neural networks under input perturbations. Specifically, for a given neural network, suppose $z_i(x)$ is the activation of the $i$th layer of a neural network with input $x$. The goal of a robustness verification algorithm is to compute layer-wise lower and upper bounds of the neural network, denoted as $\underline{z}_i(x, \epsilon)$ and $\overline{z}_i(x, \epsilon)$, such that $\underline{z}_i(x, \epsilon) \leq z_i(x + \delta) \leq \overline{z}_i(x, \epsilon)$, for any additive input perturbation $\delta$ on $x$ with $||\delta||_p \leq \epsilon$. We will apply robustness verification algorithms on the Q-networks (for DQN) or policy networks (for A3C and PPO) to get layer-wise output bounds of $Q$ and $\pi$. These output bounds can be used to calculate an upper bound of the original loss function under worst-case adversarial perturbation $\mathcal{L}_{\text{adv}}$ for Approach **#1**. Similarly, the layer-wise bound is used to minimize the overlap of output intervals as proposed in Approach **#2**. For the purpose of training efficiency, IBP [14] is used to compute the layer-wise bounds for the neural networks, but other differentiable certification methods [12, 13, 11, 31, 32, 33] could be applied directly (albeit may incur additional computation cost). Our experiments focus on $p = \infty$ to compare with baselines but the methodology works for general $p$.

## 3.1 RADIAL-DQN

For Dueling-DQN, the advantage function $A_Q$ is used to decide which action to take and the value function $V_Q$ is only important for training. Hence, we only need to make $A_Q$ robust and the Q function is lower and upper bounded by $\underline{Q}(s, a, \epsilon) = V_Q(s) + \underline{A_Q}(s, a, \epsilon)$ and $\overline{Q}(s, a, \epsilon) = V_Q(s) + \overline{A_Q}(s, a, \epsilon)$ with $\epsilon$-bounded perturbations to input $s$. In **RADIAL** we proposde two approaches to derive the adversarial loss $\mathcal{L}_{\text{adv}}$; however, due to the space constraints, we describe the approach that has better empirical performance in the main text, and leave the other in the Appendix.

For DQN, we find that Approach **#2** performs better. The goal of Approach **#2** is to minimize the weighted overlap of activation bounds for different actions (Fig. 2). The idea is to minimize only what is necessary for robust performance, *overlap*. If there is no *overlap*, the original action's Q-value is guaranteed to be higher than others even under perturbation, so the agent won't change it's behavior under perturbation. However not all overlap is equally important. If two actions have a very similar Q-value, *overlap* is acceptable, as taking a different but equally good action under perturbation is not a problem. To address this we added weighting by $Q_{\text{diff}}$, which helps by multiplying overlaps with similar Q-values by a small number and overlaps with different Q-values by a large number.

The final loss loss function is as follows:

$$\mathcal{L}_{\text{adv}}(\theta_{\text{actor}}, \epsilon) = \mathbb{E}_{(s,a,s',r)}[\sum_y Q_{\text{diff}}(s, y) \cdot Ovl(s, y, \epsilon)] \tag{5}$$

where

$$Q_{\text{diff}}(s, y) = \max(0, Q(s, a) - Q(s, y)), \quad Ovl(s, y, \epsilon) = \max(0, \overline{Q}(s, y, \epsilon) - \underline{Q}(s, a, \epsilon) + \eta)$$

$\eta = 0.5 \cdot Q_{\text{diff}}(s, y)$ and $a$ is the action taken. Here $Ovl$ represents the overlap between the bounds of two actions (grey region in Fig 2). To promote additional robustness, the network is incentivized to have a margin $\eta$ (rather than simply no overlaps). We set $\eta = 0.5 \cdot Q_{\text{diff}}$ to have it be half of the maximum margin attainable. See Appendix F for experiments and discussion on the importance of this choice for margin. Note that $Q_{\text{diff}}$ is treated as a constant (no gradient) for the optimization. Eq. (5) reduces to zero when $\epsilon = 0$.

## 3.2 RADIAL-A3C

In A3C, as the value network $V$ and entropy $\mathcal{H}$ are only used to help training, we will use the unperturbed form of the approximated advantage $A_t(s_t, a_t; \theta, \theta_v)$ and entropy $\mathcal{H}$ and focus on designing a robust policy network $\pi$ in **RADIAL**-A3C. As the Approach **#2** is more effective in our experiments, we focus on describing Approach **#2** here and leave Approach **#1** in the Appendix B. The idea of Approach #2 for A3C is very similar to DQN, except that the Q-values are replaced by a combination of the policy outputs $\pi$ and the pen-ultimate layer of the policy networks $z$ (before the softmax layer):

$$\mathcal{L}_{\text{adv}}(\theta_{\text{actor}}, \epsilon) = \mathbb{E}_{(s,a,s',r)}[\sum_y \pi_{\text{diff}}(s, y) \cdot Ovl(s, y, \epsilon)] \tag{6}$$

We design $L_{adv}$ to try to minimize the overlap between actions

Figure 2: Visualizing $L_{\text{adv}}$ for **RADIAL**-DQN (Approach #2) in a simple case with 2 actions.

---

**Algorithm 1:** *Greedy Worst-Case Reward*

---

$R = 0$
**while** $s_t$ *not terminal* **do**
    1. Calculate $\pi_i(s_t, \theta)$, $\underline{\pi}_i(s_t, \epsilon; \theta)$ and $\overline{\pi}_i(s_t, \epsilon; \theta)$ for each action $i$
    2. Calculate set of possible actions $\Gamma := \{i \mid \overline{\pi}_i \geq \max_j(\underline{\pi}_j)\}$
    3. Take the "worst" action $k$ out of the possible actions, $k = \text{argmin}_{i \in \Gamma}(\pi_i(s_t, \theta))$. Observe $r_t$ and $s_{t+1}$ and update $R \leftarrow R + r_t$
    4. $t = t + 1$
**end**
**return** $R$

---

with $\pi_{\text{diff}}(s, y) = \max(0, \pi(s, a) - \pi(s, y))$ and $Ovl(s, y, \epsilon) = \max(0, \overline{z}(s, y, \epsilon) - \underline{z}(s, a, \epsilon) + \eta)$. Where $\eta = 0.5 \cdot z_{\text{diff}}(s, y)$ and $z_{\text{diff}}(s, y) = \max(0, z(s, a) - z(s, y))$.

## 3.3 RADIAL-PPO

For **RADIAL**-PPO, we use Approach #1 as Approach #2 does not work for continuous actions(discussion in Appendix C). In **RADIAL**-PPO, we only calculate bounds over current policy output, because the experiences are generated by sampling from unperturbed old policy, and value function/advantage are only used during training. The intuition for Approach #1 is to derive an upper bound of the RHS of Eqn. 3 which will always hold.

$\forall s_t \in \mathcal{S}, \forall ||\delta||_p \leq \epsilon$ ; $\mathcal{L}(\theta, s_t + \delta) \leq \mathcal{L}_{\text{adv}}(\theta, s_t, \epsilon)$. Minimizing this upper bound will then also decrease the loss under perturbation. In practice optimizing this loss function results in increasing the lower bound of the probability of taking good actions ($A_t \geq 0$), and decreasing the upper bound of the probability to take bad actions ($A_t < 0$).

The mathematical definition of $\mathcal{L}_{\text{adv}}$ is the following:

$$\mathcal{L}_{\text{adv}}(\theta, \epsilon) = \mathbb{E}_{(s_t, a_t, r_t)} \left[ -\min(\frac{\hat{\pi}(a_t|s_t, \epsilon; \theta)}{\pi(a_t|s_t; \theta_{\text{old}})} A_t, \text{clip}(\frac{\hat{\pi}(a_t|s_t, \epsilon; \theta)}{\pi(a_t|s_t; \theta_{\text{old}})}, 1 - \eta, 1 + \eta) A_t) \right] \quad (7)$$

The worst-case policy is defined as:

$$\hat{\pi}(a_t|s_t, \epsilon; \theta) = \begin{cases} \underline{\pi}(a_t|s_t, \epsilon; \theta), & \text{if } A_t \geq 0 \\ \overline{\pi}(a_t|s_t, \epsilon; \theta), & \text{otherwise} \end{cases} \quad (8)$$

For our continuous control experiments, the output of our policy are the parameters $\mu, \Sigma$ of a Gaussian, with covariance being diagonal and independent of input state $s$. We can define bounds on distance $d$:

$\overline{d}(a_t, s_t, \epsilon) = \max_{\mu \in \{\underline{\mu}, \overline{\mu}\}} (a_t - \mu)^T \Sigma^{-1}(a_t - \mu)$, $\underline{d}(a_t, s_t, \epsilon) = \min_{\mu \in [\underline{\mu}, \overline{\mu}]} (a_t - \mu)^T \Sigma^{-1}(a_t - \mu)$

Then $\overline{\pi}(a_t|s_t, \epsilon; \theta) = \frac{e^{-\underline{d}/2}}{((2\pi)^k \det \Sigma)^{0.5}}$ and $\underline{\pi}(a_t|s_t, \epsilon; \theta) = \frac{e^{-\overline{d}/2}}{((2\pi)^k \det \Sigma)^{0.5}}$. Where $k$ is the number of dimensions in the action space.

For discrete action, $\pi$ is a categorical distribution over possible actions, where $\overline{\pi}(a_t)$ is the upper bound of the policy network $\pi$ at $a_t$-th output. This can be computed from the upper bound of $a_t$-th logit(output before softmax) and lower bound of other logits by applying softmax.

## 3.4 New efficient evaluation metric: Greedy Worst-Case Reward

The goal of training RL agents robust against input perturbations is to ensure that the agents could still perform well under any (bounded) adversarial perturbations. This can be translated into maximizing the *worst-case* reward $R_{wc}$, which is the reward under worst possible sequence of adversarial attacks.

We define $R_{wc}$ as follows: $R_{wc} = \min_{||\delta_t||_p \leq \epsilon} \mathbb{E}_\tau[R(\tau)]$ with trajectory $\tau = (s_0, a_0, ..., s_T, a_T)$ where $a_t, s_t, r_t$ are drawn from $\pi(s_t + \delta_t), \mathcal{P}(s_{t-1}, a_{t-1}), \mathcal{R}(s_{t-1}, a_{t-1})$ respectively, and $R(\tau) = \sum_t r_t$. One idea is to evaluate every possible trajectory $\tau$ to find which one produces the minimal reward. However $R_{wc}$ is practically impossible to evaluate, as finding the worst perturbations $\delta_t$ of a set of possible actions $a_t$ for each state $s_t$ is NP-hard, and the amount of trajectories to evaluate grows exponentially with respect to trajectory length $T$, which is hard to compute. One possible way to avoid finding worst-case perturbations directly is to use certified output bounds [12, 13, 11, 31, 32, 33], which produces a superset of all possible actions under worst-case perturbations and hence the resulting total accumulative reward is a lower bound of $R_{wc}$. We name this reward as Absolute Worst-Case Reward (AWC). Note that AWC is a lower bound of $R_{wc}$ when both the policy and environment are deterministic.

However, AWC still requires evaluating an exponential amount of possible action sequences, which is computationally expensive. To overcome this limitation, we propose an alternative evaluation method called *Greedy Worst-Case Reward* (GWC) in Algorithm 1, which approximates the desired $R_{wc}$ and can be computed efficiently with a *linear* complexity of total timesteps $T$. The idea of GWC is to avoid evaluating exponential numbers of trajectories and use a simple heuristic to approximate $R_{wc}$ by choosing the action with lowest Q-value (or the probability of action taken for A3C) greedily at each state. We show in Fig 4 (in Appendix E) that GWC is often close to AWC while being much faster to evaluate (*linear* complexity of total time steps). Discussion about metrics used in baseline works [17, 18] and full description of the algorithm for computing AWC is in Appendix D.

## 4 Experimental results

**Environments and setup** To have a fair comparison with baseline works [17, 18], we experiment on the same Atari-2600 environment [35] and same 4 games used in [17, 18]. Different from [17, 18], we further evaluate our algorithm on a more challenging ProcGen [36] benchmark, which allows us to test generalization abilities of the agent. Note that both Atari-games and ProcGen [36] benchmark have high dimensional pixel inputs and discrete action spaces. To compare our RL agents with continuous actions with [17], we use the MuJoCo environment which simulates robotic control and has relatively low-dimensional inputs and a continuous action space. Full training details and hyperparameter settings are available in Appendix H.

**Evaluation.** We evaluate the performance of our agents with a total of 3 metrics: (a) total reward under 10-steps $l_\infty$-PGD untargeted attack applied on every frame, (b) GWC and (c) Action Certification Rate (ACR)[17]. For Atari games the result of Approach #2 for **RADIAL**-agents are in Table 1, and the results of Approach #1 are in Appendix I. Note that A3C and PPO take actions stochastically during training but we set them to deterministically choose the action that has the highest probability during evaluation.

### 4.1 Atari results

Table 1 shows that **RADIAL**-DQN outperforms or matches all the baselines on all four games against $\epsilon = 1/255$ PGD-attacks with the same evaluation method in [17]. The result suggests that **RADIAL** can train a robust policy against $\epsilon = 1/255$ adversarial perturbations without sacrificing nominal performance. In fact, our robust **RADIAL** agents consistently outperform the baselines on all the evaluation metrics, with significant margin in RoadRunner. In addition to better rewards, our **RADIAL**-DQN is roughly $6\times$ faster to train than [17]. Experimentally if we include the standard training process, our total run time is only 17 hours compared to SA-DQN's 35 hours on our hardware. **RADIAL**-A3C also achieves high rewards under adversarial attacks and beats the baselines on 2/4 tasks despite the games likely being chosen by previous work because they are easy for Q-learning agents. As shown in Table 1 (A3C is excluded because it did not learn Freeway), **RADIAL**-A3C reaches slightly lower nominal rewards than **RADIAL**-DQN but shows more robust performance against large perturbations, even outperforming **RADIAL**-DQN against $\epsilon = 5/255$ PGD attacks on RoadRunner. This shows **RADIAL** works well for very different types of RL algorithms.

One interesting finding is that our A3C baseline actually performed well against small PGD-attacks even without robust training (the standard DQN models all perform terribly at $\epsilon = 1/255$ while A3C has comparable or even better performance than the robust trained SA-DQN or RS-DQN.) We believe this is due to two main factors: (a) the network architecture is slightly different from DQN

| | Model/metric | Nominal | | PGD attack | | | GWC | ACR |
|---|---|---|---|---|---|---|---|---|
| | $\epsilon$ | 0 | 1/255 | 3/255 | 5/255 | 1/255 | 1/255 |
| *BankHeist* **Baselines:** | | | | | | | |
| Standard | DQN [17] | 1325.5±5.7 | 29.5±2.4 | 0.0±0.0 | 0.0±0.0 | 0.0±0.0 | 0.000 |
| | A3C | 1109.0±21.4 | 1102.5±49.4 | 534.5±58.2 | 115.0±27.8 | 0.5±0.5 | 0.000 |
| Robust | RS-DQN [18] | 238.66 | 190.67 | N/A | N/A | N/A | N/A |
| | SA-DQN [17] | 1237.6±1.7 | 1237.0±2.0 | 1213.0±2.5 | 1130.0±29.1 | 1196.5±9.4 | 0.976 |
| **Our Methods:** | | | | | | | |
| | RADIAL-DQN | **1349.5±1.7** | **1349.5±1.7** | **1348±1.7** | **1182.5±43.3** | **1344.5±1.8** | 0.981 |
| | RADIAL-A3C | 1036.5±23.4 | 975±22.2 | 949±19.5 | 712±46.4 | 851.5±2.9 | 0.718 |
| *Freeway* **Baselines:** | | | | | | | |
| Standard | DQN [17] | **33.9±0.07** | 0.0±0.0 | 0.0±0.0 | 0.0±0.0 | 0.0±0.0 | 0.000 |
| Robust | RS-DQN [18] | 32.93 | 32.53 | N/A | N/A | N/A | N/A |
| | SA-DQN [17] | 30.0±0.0 | 30.0±0.0 | 30.05±0.05 | 27.65±0.22 | 30.0±0.0 | 1.000 |
| **Our Methods:** | | | | | | | |
| | RADIAL-DQN | 33.2±0.19 | **33.35±0.16** | **33.4±0.13** | **29.1±0.17** | **33.25±0.24** | 0.998 |
| *Pong* **Baselines:** | | | | | | | |
| Standard | DQN [17] | **21.0±0.0** | -21.0±0.0 | -21.0±0.0 | -20.85±0.08 | -21.0±0.0 | 0.000 |
| | A3C | **21.0±0.0** | **21.0±0.0** | **21.0±0.0** | -17.85±0.33 | -21.0±0.0 | 0.000 |
| Robust | RS-DQN [18] | 19.73 | 18.13 | N/A | N/A | N/A | N/A |
| | SA-DQN [17] | **21.0±0.0** | **21.0±0.0** | **21.0±0.0** | -19.75±0.1 | **21.0±0.0** | 1.000 |
| **Our Methods:** | | | | | | | |
| | RADIAL-DQN | **21.0±0.0** | **21.0±0.0** | **21.0±0.0** | **21.0±0.0** | **21.0±0.0** | 0.894 |
| | RADIAL-A3C | **21.0±0.0** | **21.0±0.0** | **21.0±0.0** | **21.0±0.0** | **21.0±0.0** | 0.755 |
| *RoadRunner* **Baselines:** | | | | | | | |
| Standard | DQN [17] | 43390±973 | 0.0±0.0 | 0.0±0.0 | 0.0±0.0 | 0.0±0.0 | 0.000 |
| | A3C | 34420±604 | 31040±2173 | 3025±317 | 350±93 | 0.0±0.0 | 0.000 |
| Robust | RS-DQN [18] | 12106.67 | 5753.33 | N/A | N/A | N/A | N/A |
| | SA-DQN [17] | **45870±1380** | 44300±1753 | 20170±1822 | 3350±335 | 0.0±0.0 | 0.602 |
| **Our Methods:** | | | | | | | |
| | RADIAL-DQN | 44495±1165 | **44445±1148** | 39560±1621 | 23820±942 | **45770±1622** | 0.994 |
| | RADIAL-A3C | 34825±981 | 31960±933 | 29920±1496 | **31545±1480** | 31885±1912 | 0.923 |

Table 1: We report the mean reward of 20 runs as well as the standard error of the mean(sem). Best result boldfaced, results within a sem of the best result underlined. RADIAL-DQN outperforms all the baselines. All the robust models are trained with $\epsilon = 1/255$. RS-DQN results are N/A besides 1/255 PGD as the authors have not released code or models.

and uses Max-pooling layers, which naturally makes it more resistant against many small changes to input; (b) A3C training is more stochastic, so it is more resistant to occasional random actions caused by the untargeted attack.

Another interesting observation in our experiments across different algorithms and environments is that sometimes applying an attack or increasing the magnitude of PGD-attack increases the reward achieved. We note that this phenomenon is not unreasonable since the PGD attack is designed to simply change original actions of RL agent; it is possible and often even likely that original trajectory was not the best possible one and a different trajectory can give higher reward. We observed this phenomenon on Freeway for both SA-DQN and **RADIAL**-DQN (Table 1), and for **RADIAL**-PPO on CoinRun and Jumper (Table 2). We believe this indicates the weakness of untargeted PGD-attacks and highlights room for improvement of attacks on RL-policies.

## 4.2 ProcGen Results

We report our results on 3 environments on the ProcGen benchmark [36] in Table 2. To our best knowledge, it is the first evaluation of robustness of RL-agents trained on ProcGen. All our agents were trained for 25M steps on the easy setting, trained on 200 different levels and evaluated on the

| Env: | Model | Dist | Nominal | $\epsilon$=1/255 PGD | $\epsilon$=3/255 PGD | $\epsilon$=5/255 PGD |
|------|-------|------|---------|------------|------------|------------|
| Fruitbot | PPO | Train | **30.20 $\pm$ 0.23** | 25.72 $\pm$ 0.33 | 15.56 $\pm$ 0.38 | 8.79 $\pm$ 0.35 |
| | | Eval | **26.09 $\pm$ 0.33** | 22.53 $\pm$ 0.38 | 13.51 $\pm$ 0.39 | 8.51 $\pm$ 0.35 |
| | **RADIAL**-PPO | Train | 28.03 $\pm$ 0.24 | **27.93 $\pm$ 0.25** | **27.84 $\pm$ 0.26** | **27.50 $\pm$ 0.26** |
| | | Eval | 26.08 $\pm$ 0.29 | **26.06 $\pm$ 0.30** | **25.87 $\pm$ 0.30** | **25.81 $\pm$ 0.30** |
| Coinrun | PPO | Train | **8.31 $\pm$ 0.12** | 6.36 $\pm$ 0.15 | 4.19 $\pm$ 0.16 | 3.32 $\pm$ 0.15 |
| | | Eval | 6.65 $\pm$ 0.15 | 5.22 $\pm$ 0.16 | 3.58 $\pm$ 0.15 | 3.36 $\pm$ 0.15 |
| | **RADIAL**-PPO | Train | 7.12 $\pm$ 0.14 | **7.10 $\pm$ 0.14** | **7.19 $\pm$ 0.14** | **7.34 $\pm$ 0.14** |
| | | Eval | **6.66 $\pm$ 0.15** | **6.71 $\pm$ 0.15** | **6.71 $\pm$ 0.15** | **6.67 $\pm$ 0.15** |
| Jumper | PPO | Train | **8.69 $\pm$ 0.11** | 6.61 $\pm$ 0.15 | 4.50 $\pm$ 0.16 | 3.42 $\pm$ 0.15 |
| | | Eval | **4.22 $\pm$ 0.16** | 3.90 $\pm$ 0.15 | 3.10 $\pm$ 0.15 | 3.15 $\pm$ 0.15 |
| | **RADIAL**-PPO | Train | 6.59 $\pm$ 0.15 | **6.70 $\pm$ 0.15** | **6.55 $\pm$ 0.15** | **6.83 $\pm$ 0.15** |
| | | Eval | 3.85 $\pm$ 0.15 | **3.93 $\pm$ 0.15** | **3.75 $\pm$ 0.15** | **3.59 $\pm$ 0.15** |

Table 2: Results on the ProcGen environments. Each model was evaluated for 1000 episodes on the training/evaluation set. Reported means together with standard error of the mean.

| Environment | Model | Nominal $\epsilon = 0$ | $\epsilon = 0.0375$ | Attack (MAD) $\epsilon = 0.075$ | $\epsilon = 0.15$ |
|-------------|-------|---------|-----------|---------|---------|
| Walker2D | PPO | 3635.3$\pm$50.7 | 1968.5$\pm$90.6 | 1283.1$\pm$61.9 | 670.3$\pm$31.9 |
| ($\epsilon_{\text{train}} = 0.075$) | SA-PPO [17] | 3776.4$\pm$186.0 | 3923.6$\pm$189.9 | 3263.7$\pm$228.1 | 3652.1$\pm$203.3 |
| | **RADIAL**-PPO | **5251.6$\pm$10.4** | **5108.4$\pm$67.9** | **4474.7$\pm$140.6** | **3895.3$\pm$128.3** |
| Hopper | PPO | 2740.7$\pm$125.8 | 2034.4$\pm$126.9 | 1524.2$\pm$146.3 | 969.9$\pm$90.1 |
| ($\epsilon_{\text{train}} = 0.075$) | SA-PPO [17] | 3585.0$\pm$56.4 | 3364.7$\pm$93.9 | 3165.1$\pm$107.1 | 2248.3$\pm$124.7 |
| | **RADIAL**-PPO | **3737.5$\pm$4.4** | **3684.9$\pm$27.4** | **3252.1$\pm$101.9** | **2498.9$\pm$130.0** |
| Half Cheetah | PPO | **5566.0$\pm$8.7** | **5517.8$\pm$8.8** | **5179.3$\pm$64.7** | 1483.6$\pm$193.4 |
| ($\epsilon_{\text{train}} = 0.15$) | SA-PPO [17] | 4175.0$\pm$29.9 | 4168.8$\pm$29.9 | 4178.5$\pm$35.3 | **4173.1$\pm$31.8** |
| | **RADIAL**-PPO | 4724.3$\pm$10.6 | 4629.7$\pm$36.9 | 4480.0$\pm$66.9 | 4008.5$\pm$119.5 |

Table 3: We report the results of reward (mean $\pm$ standard error of the mean) for each agent in MuJoCo continuous control tasks. Agents are trained for 4096k steps and evaluated for 50 episodes on the evaluation set. Results within standard error of the mean from best result underlined.

full distribution. We used the IMPALA-CNN architecture [36], which is a significantly larger than the ones in our Atari experiments, but our algorithm and training setup can still successfully scale to this more complex setting. We notice like A3C, the original PPO-policy is already quite robust against small PGD-attacks, but **RADIAL**-PPO has even better robustness to much stronger attacks. Next, we use the train/evaluation split in levels to study whether our robust training generalizes to new environments. We observe that **RADIAL** training increases robustness consistently on both training and evaluation set, and the gap between training and evaluation results is often smaller for **RADIAL**-PPO. The results in Table 2 uses a deterministic version of the policy and the result of stochastic policy is reported in Appendix I.

## 4.3  MuJoCo Results

In Table 3 we show our results on 3 environments from MuJoCo [37] and comparison with vanilla PPO method and SA-PPO [17] convex relaxation approach. We directly use the implementation from [17]. We train 4096 k steps for all 3 environments, whereas [17] uses roughly 2000k for Walker2D and Hopper. For each environment, we compare the performance under MAD attacks [17] with different $\epsilon$. For Walker2D, we use training $\epsilon = 0.075$ instead of $\epsilon = 0.05$ from [17] for better comparison under different attacks. For Walker2D and Hopper, **RADIAL**-PPO outperforms SA-PPO [17] in both natural testing and robust testing. Interesting, for Half-Cheetah, we found this environment is naturally robust (even better than robust trained agents) against attack up to $\epsilon$=0.075.

## 4.4 Evaluating GWC

We show empirically that GWC is indeed a good approximation of AWC reward. We compare GWC and action certification rate (ACR) against the Absolute Worst-Case Reward (AWC) on a small scale experiments on reaching the first reward within 80 frames on Freeway due to the exponential complexity of AWC. Figure is available in Appendix D. GWC is the ratio of +1 rewards as measured by GWC, while AWC uses depth first search to compute the percentage of +1 rewards using Algorithm 2 (i.e. where all possible sequences of actions get at least +1 reward). For the episodes that we evaluated, GWC matched AWC perfectly for $\epsilon \in \{1.0, 1.2, 1.5\}$. Since AWC is a lower bound of GWC, for $\epsilon \in \{1.3, 1.4\}$ GWC matched AWC on 37/40 episodes, while overestimating on the 3/40. ACR (as used in [17]) did not show as strong of a correlation with AWC. As evident in Table 1, GWC is a better indicator for the robust performance than ACR – this is the case for Freeway and Pong where SA-DQN has higher action certification rates but **RADIAL**-DQN has higher GWC and PGD rewards. Together these results give faith in GWC being a good evaluation method.

## 5 Conclusions and Future works

We have shown that using the proposed **RADIAL** framework can significantly improve the robustness of deep RL agents against adversarial perturbations based on robustness verification bounds – our robustly trained agents reach very good performance against even $5\times$ larger perturbations than previous state of the art, and in the meantime are much more computationally efficient to train. In addition, we have presented a new evaluation method, *Greedy Worst-Case Reward*, as a good surrogate of the worst-case reward to evaluate deep RL agent's performance under adversarial input perturbations. Future works include extending our framework to defend against semantic perturbations such as contrast and brightness changes, which are more realistic on the vision-based RL agents (e.g. self-driving car, vision-based robots).

## 6 Limitations and Potential negative impact

An important limitation of our work and one that may cause negative impact in the future is that our work addresses only a specific axis of robustness, robustness against $l_p$-norm bounded perturbations. While there are ways to extend this, for example to semantic perturbation [28], other axes like robustness to changing environment conditions are not addressed by our method. We want to highlight common terms used in this field like certified defense and robustness may sound very convincing to a non-expert, but insufficient understanding of the limits of current robust training methods and overly relying on them presents pressing potential for negative social impact.

## Acknowledgement

The authors would like to thank anonymous reviewers for valuable feedback to improve the manuscript. The authors thank MIT-IBM Watson AI lab and the MIT SuperUROP program for support in this work. T. Oikarinen and T.-W. Weng are supported by National Science Foundation under Grant No. 2107189.

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
