# A   Appendix: Radial-DQN

**Approach #1.** A strict upper bound of the perturbed standard loss, $\max_{||\delta||_p \leq \epsilon} \mathcal{L}_{\text{nom}}$, can be constructed below:

$$\mathcal{L}_{\text{adv}}(\theta, \epsilon) = \mathbb{E}_{(s,a,s',r)}\big[\max(B_l(a), B_u(a)) + \sum_{y \neq a} \max(C_l(y), C_u(y))\big] \tag{9}$$

where we define $B = r + \gamma \max_{a'} Q_{target}(s', a')$, $B_l = (B - \underline{Q}_{actor}(s, y, \epsilon))^2$, $B_u = (B - \overline{Q}_{actor}(s, y, \epsilon))^2$, and $C_l = (Q_{actor}(s, y) - \underline{Q}_{actor}(s, y, \epsilon))^2$, $C_u = (Q_{actor}(s, y) - \overline{Q}_{actor}(s, y, \epsilon))^2$. The $B$ terms describe an upper bound of the original DQN loss function under adversarial perturbations, and the $C$ terms ensure the bounds on actions not taken will also be tight. Note that Eq. (9) reduces to the original loss Eq. (1) when $\epsilon = 0$.

# B   Appendix: Radial-A3C

**Approach #1.** Here we define the corresponding $\mathcal{L}_{adv}$ as follows to make it an upper bound of the original loss (Eq. (2)) under worst-case adversarial input perturbations:

$$\mathcal{L}_{\text{adv}}(\theta, \theta_v) = \mathbb{E}_{(s_t, a_t, r_t)}\big[A^2 - D - \beta \mathcal{H}(\pi(\theta))\big], \tag{10}$$

where

$$D = \begin{cases} \log(\underline{\pi}(a_t | s_t, \epsilon; \theta))A, & \text{if } A \geq 0, \\ \log(\overline{\pi}(a_t | s_t, \epsilon; \theta))A, & \text{otherwise.} \end{cases}$$

Here $\overline{\pi}(a_t)$ is the upper bound of the policy network $\pi$ at $a_t$-th output, which can be computed from the upper bound of $a_t$-th logit and lower bound of other logits due to the softmax function at the last layer. $\mathcal{L}_{adv}$ is an upper bound and reduces to $\mathcal{L}_{standard}$ if $\epsilon = 0$.

# C   On Approach #2 for continuous actions

In Approach #2, we design $\mathcal{L}_{adv}$ to minimize the overlap between lower bound of the chosen action $a$ and upper bounds of all other actions $y \in \mathcal{A} \setminus \{a\}$. This approach inherently relies on the action space $\mathcal{A}$ being discrete. If we wanted to extend this idea to continuous action, we could replace the summation of Eqn. 6 with an integral over $\mathcal{A} \setminus \{a\}$. Since action $a$ is a single point on the action space $\mathcal{A}$, the integral over $\mathcal{A} \setminus \{a\}$ is same as integral over the whole action space. This is not desirable as $a$ will always overlap with itself and this is not something we wish to regularize. In addition, calculating such an integral will not be feasible in general and would have to be approximated by Monte Carlo sampling from uniform distribution $\mathcal{A}$.

Even if we overcome the integration issues, we are still faced with challenges defining the overlap term. For continuous control the network does not give a separate output for each possible action, which means there's no action specific upper/lower bounds needed to calculate Overlap. One possibility would be to use something like $Ovl(s, y, \epsilon) = \max(0, \overline{\pi}(s, y, \epsilon) - \underline{\pi}(s, a, \epsilon) + \eta)$, but since the action probabilities are proportional to distance from the mean of the output, we believe it would be better to simply use $\mathcal{L}_{adv}$ explicitly designed for continuous control.

# D   Appendix: Discussion on GWC and the metrics used in [17, 18]

Existing evaluation methods in previous works do not directly measure reward. For example, [17] uses the action certification rate and [18] uses the size of average provable region of no action change. These evaluation methods primarily focus on not changing agent's original actions under adversarial perturbations, which can be useful. When most actions don't change under attack, the reward is also less likely to change. However, this is often not enough as attacks changing just one early action could push the agent to an entirely different trajectory with very different results. As such, high action certification rates may not result in a high reward.

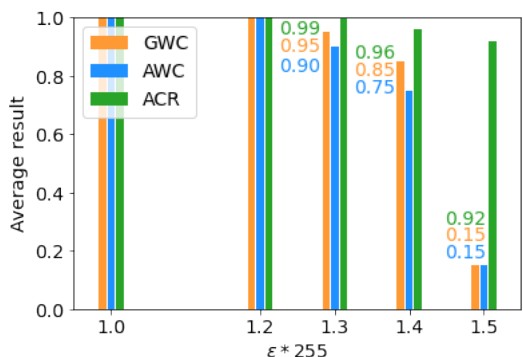

Figure 3: The means over 20 runs of RADIAL-DQN model evaluated on first 80 steps of Freeway games with different perturbation sizes. GWC is the percentage episodes that reach +1 reward within first 80 steps of an episode measured by *Greedy Worst-Case Reward* , while AWC is the percentage of +1 rewards using absolute worst-case calculation.

To showcase this we have presented a comparison of GWC, AWC, and ACR in Figure 3, described in more detail in Section 4.4. In addition a full description of AWC is provided in Algorithm 2.

---

**Algorithm 2:** Absolute Worst-Case Reward

---

$S_{open} = \{(s_0, 0)\}$ and $R_{min} = \infty$
**while** $S_{open} \neq \emptyset$ **do**
    1. Pick a state and reward tuple $(s', R')$ from $S_{open}$ and remove it from the set.
    2. Calculate $\pi_i(s', \theta)$, and $\underline{\pi}_i(s', \epsilon; \theta), \overline{\pi}_i(s', \epsilon; \theta)$ for each action $i$
    3. Calculate set of possible actions $\Gamma := \{i \mid \overline{\pi}_i \geq \max_j(\underline{\pi}_j)\}$
    4. **for** *action i in* $\Gamma$ **do**
        Take action $i$, and observe new state $s''$ and reward $r''$.
        **if** $s''$ *is terminal* **then**
            | Update $R_{min} \leftarrow \min(R_{min}, R' + r'')$
        **end**
        **else**
            | Add $(s'', R' + r'')$ to $S_{open}$
        **end**
    **end**
**end**
**return** $R_{min}$

---

# E Appendix: Q-value difference

## E.1 Atari results

One of the main differences between our **RADIAL**-DQN Approach #2 and SA-DQN is that our formulation does not cause a bias in the Q-values of the network. This is done by requiring the gap between the output bounds of two actions to be half of the distance between their Q-values, whereas SA-DQN requires this to be 1 – this can cause issues when the natural Q-values differ by less than 1. To achieve such a large gap, the networks needs to increase the higher Q-value and decrease the lower one. To support our argument, Figure 4 plots the errors in Q-value of both SA-DQN and RADIAL-DQN, which is defined as (predicted Q-value) - (ideal Q-value), with ideal Q-value being the cumulative time discounted reward of the rest of the episode. It shows that SA-DQN does have higher bias in Q-values than ours, which is an undesired effect and potentially problematic.

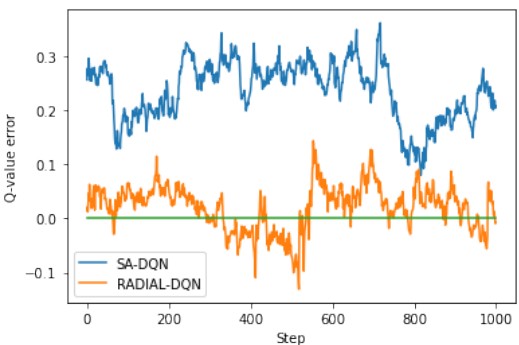

Figure 4: SA-DQN consistently overestimates its Q-values on an episode of Freeway.

|  | Nominal | 1/255 PGD | 3/255 PGD | 5/255 PGD |
|---|---|---|---|---|
| **BankHeist** | | | | |
| c=0.25 | 1347.5 | 1214.5 | 1257 | 1113.5 |
| c=0.5(default) | **1349.5** | **1349.5** | **1348** | 1182.5 |
| c=0.75 | 896 | 896 | 896 | 896 |
| Symmetric loss(c=0.5) | 1325 | 1325 | 1325 | **1305** |
| **RoadRunner** | | | | |
| c=0.25 | 20385 | 15135 | 10795 | 11220 |
| c=0.5(default) | **44495** | **44445** | **39560** | **23820** |
| c=0.75 | 4870 | 4870 | 4870 | 4870 |
| Symmetric loss(c=0.5) | 34905 | 35315 | 33315 | 21385 |

Table 4: Experiment on changing margin coefficient or a more symmetric loss function for RADIAL-DQN on Atari.

## F   Appendix: Alternative Approach 2 loss formulations

### F.1   Margin

The main idea to choose the margin $\eta$ in Equation 5 is as follows: the margin should be $c \cdot Q_{\text{diff}}$ with $0 < c < 1$. If Margin=$Q_{\text{diff}}$, it is equal to the situation where the bounds are exactly tight (for example if $\epsilon = 0$), and thus it's impossible to have a margin bigger than $Q_{\text{diff}}$. We did notice that it is important to require a margin $c > 0$ as our experiments with $c = 0$ did not reach desired robustness. We initially used $c = 0.5$ for simplicity, and did not tune as it worked well in our experiments. Note that SA-DQN[38] uses a constant margin requirement in a loss function somewhat similar to ours. This has undesirable effects when $Q_{\text{diff}}$ is smaller than this margin, see Appendix E for more discussion on this.

To understand the sensitivity of our algorithm to this choice of $c$, we have conducted additional experiments on the two games Roadrunner and BankHeist . We observed that the choice of $c$ does affect the robust agent's performance, significantly in some games such as RoadRunner. Full results in the table 4.

We can see that $c = 0.75$ produces poor standard performance and we think it's because this requirement is too strict and the policy collapses to some simple policy with lower reward. For $c = 0.25$, the resulting robust policy works well and has only slightly worse robustness than default in BankHeist; however, for RoadRunner, the results are much worse than with $c = 0.5$.

### F.2   Symmetric Loss Function

In the current loss function(Eqn. 5), if we take the action with the lowest Q-value by chance, the $\mathcal{L}_{\text{adv}}$ term simply goes to zero, which may not be desirable. Inspired by reviewer suggestion, we have

| MuJoCo Environment | Nominal | MAD $\epsilon = 0.075$ | Compund attack, $\epsilon = 0.075$ |
|---|---|---|---|
| Walker2D | $5251.6 \pm 10.4$ | $4474.7 \pm 140.6$ | $4349.9 \pm 127.0$ |
| Hopper | $3737.5 \pm 4.5$ | $3252.1 \pm 101.9$ | $3439.1 \pm 70.3$ |
| Half Cheetah | $4724.3 \pm 10.6$ | $4480.0 \pm 66.9$ | $4382.5 \pm 142.8$ |

Table 5: Compounding attack experiments on MuJoCo agents trained by RADIAL-PPO

conducted the following experiments to design a more symmetric version of the loss function Eqn. 5 to see if it can increase RADIAL-DQN performance.

The new loss is as follows: we added a second term inside the summation of Eqn 5 with terms flipped to have the same regularization when Q-value for $a$ is lower than $y$, i.e. $Q'_{diff} = \max(0, Q(s, y) - Q(s, a))$ and $Ovl' = \max(0, \underline{Q}(s, y, \epsilon) - \overline{Q}(s, a, \epsilon) + 0.5 \cdot Q'_{diff})$. The full loss is then

$$\mathcal{L}_{\text{adv}}(\theta_{\text{actor}}, \epsilon) = \mathbb{E}_{(s,a,s',r)}[\sum_{y} Q_{\text{diff}}(s, y) \cdot Ovl(s, y, \epsilon) + Q'_{\text{diff}}(s, y) \cdot Ovl'(s, y, \epsilon)]$$

We tested this new loss on BankHeist and RoadRunner for Atari. For BankHeist, it's slightly worse nominal performance than our default RADIAL-DQN, but a little better performance against 5/255 PGD. Full results are summarized in below Table 4.

We hypothesize that the slightly worse performance could be caused by more strict enforcement of the margin between outputs in this new loss, which could potentially make the policy harder to correct if wrong Q-values are reached at some point.

# G   Compounding attack on MuJoCo

To test whether our defense still performs well against compounding attacks, we have conducted a small experiment using the compounding attack described in [8] against our trained RADIAL-PPO models on MuJoCo.

Here is a brief description of our approach to compounding attack:

1. Given the MuJoCo environment, learn a model $F$ for the dynamical system where $F$ approximates the next state given current state and action, $s_{i+1} \approx F(s_i, a_i)$

2. For a forward step $i$ in testing, given learned policy model $\pi$ from RADIAL-PPO training( $a_i \sim \pi(s_i)$), simulate the next n compounding steps by forward iteration $s_{i+1} \sim F(s_i, \pi(s_i)), s_{i+2} \sim F(s_{i+1}, \pi(s_{i+1}))$ ..., denote the last simulated state as $s_{target}$.

3. Our goal is to apply adversarial attack to $s_i$, such that the trajectory is deviated from original simulation as far as possible. We start another compounding loop by introducing $\delta s_i$ as perturbation, and $s_{perturbed}$ will be the last state of the iteration. By taking the gradient of $s_{perturbed} - s_{target}$ with respect to $\delta s_i$ and updating the perturbation, we can attack the observation state to have the model perform worse.

Table 5 shows our preliminary results for compounding attacks. Here the number of compounding steps is chosen as 3. We found the attack strength is similar compared with the MAD attack while there are few improvement points. First, the design of the target function can be changed. Our current approach is to deviate the trajectory, a more reasonable one could be to design a loss function fooling the agent to fail. Secondly there is room for improvement by tuning our learned dynamics model and compounding attack hyper-parameters. However this initial result increases our confidence that our current training helps defend against compounding attacks.

# H   Appendix: Training details

## H.1   Atari training details

**RADIAL-DQN & RADIAL-A3C.** For Atari games we first train a standard agent without robust training, and then *fine-tune* the model with **RADIAL** training. We found this training flow generally improve effectiveness of training and enable the agents to reach high nominal rewards. For DQN we use the same architecture as [17] and their released standard (non-robust) model as the starting point for fine-tuning to have a fair comparison – this makes sure the difference in performance is caused by the robust training procedure. For A3C we trained our own standard model.

The standard DQN was trained for 6M steps followed by 4.5M steps of **RADIAL** training. For **RADIAL**-DQN training, we used $\kappa = 0.8$, and increased attack $\epsilon$ from 0 to 1/255 during the first 4M steps with the smoothed linear epsilon schedule in [17]. For A3C, we first train A3C models for 20M steps with standard training followed by 10M steps of **RADIAL**-A3C training, requiring a similar computational cost as our DQN training. For **RADIAL**-A3C training, $\epsilon$ was increased from 0 to 1/255 over the first 2/3 of the training steps using the smoothed linear schedule and kept at 1/255 for the rest, and we set $\kappa \in \{0.8, 0.9\}$.

**DQN architechture**   The DQN architecture starts with a convolutional layer with 8x8 kernel, stride of 4 and 32 channels, followed by a convolutional layer with 4x4 kernel, stride of 2 and 64 channels, and then a convolutional layer with 3x3 kernel, stride of 1 and 64 channels. This is then flattened and fed into two separate 512 unit fully connected layers, one of which is connected to 1 unit value output, and the other is connected to the advantage outputs which has the size of the action space. Each layer (except for the output layers) is followed by nonlinear ReLU activations.

**A3C architechture**   The A3C uses the following architecture: two convolutional layer with 5x5 kernel, stride of 1 and 32 channels followed by a 2x2 maxpooling layer each, then a convolutional layer with 4x4 kernel, stride of 1 and 64 channels followed by 2x2 maxpool, next a convolutional layer with 3x3 kernel, stride of 1 and 64 channels again followed by 2x2 max pool. Finally, it is followed by a fully connected layer with 512 units, which is connected to two output layers, a 1 unit output layer for value output $V$ which has no activation function, and a policy output followed by a softmax activation. Additionally ReLU nonlinearities are applied after each maxpooling layer and the fully connected layer.

**Environment details**   All our models take an action or step every 4 frames, skipping the other frames. The network inputs were 84x84x1 crops of the grey-scaled pixels with no frame-stacking, scaled to be between 0-1. All rewards were clipped between [-1, 1].

**Computing infrastructure**   The models were trained in various settings. For the reported DQN training time, we used a system with two AMD Ryzen 9 3900X 12-Core CPUs and a GeForce RTX 2080 GPU with 8GB of memory.

**DQN hyper-parameters**   For all DQN models, we used Adam optimizer [39] with learning rate of $1.25 \cdot 10^{-4}$ and $\beta_1 = 0.9$, $\beta_2 = 0.999$. We used Dueling DQN with a replay buffer of $2 \cdot 10^5$, and $\epsilon_{exp}$-end of 0.05 for all games except 0.01 for RoadRunner. The neural network was updated with a batch-size of 128 after every 8 steps taken, and the target network was updated every 2000 steps taken.

The hyper-parameters include learning rate chosen from $\{6.25 \cdot 10^{-5}, 1.25 \cdot 10^{-4}, 2.5 \cdot 10^{-4}, 5 \cdot 10^{-4}\}$, $\epsilon_{exp}$-end from $\{0.01, 0.02, 0.05, 0.1\}$, batch size from $\{32, 64, 128, 256\}$, $\kappa$ from $\{0.5, 0.7, 0.8, 0.9, 0.95, 0.98\}$ and and were chosen based on what performed best on Pong training and kept the same for other tasks except for RoadRunner $\epsilon_{exp}$-end.

**A3C hyper-parameters**   A3C models were trained using all 16 cpu workers and 4 GPUs for gradient updates, in which setting training runs took around 4 hours for both standard and robust training. We used Amsgrad optimizer at a learning rate of 0.0001, $\beta_1 = 0.9$, $\beta_2 = 0.999$ for all A3C models. Our $\beta$ controlling entropy regularization was set to 0.01, and $k$ in advantage function to

| $\epsilon$ | 0 (nominal) | 1/255 (attack) | 5/255 (attack) |
|---|---|---|---|
| **BankHeist** | | | |
| Ours (IBP) | 1349.5±1.7 | 1349.5±1.7 | 1348±1.7 |
| Ours (C-IBP) | 1337.5±6.7 | 1337.5±6.7 | 1328.5±3.4 |
| SA-DQN (IBP) | 0.0 ± 0.0 | 0.0±0.0 | 0.0 ± 0.0 |
| **Pong** | | | |
| Ours (IBP) | 21.0±0.0 | 21.0±0.0 | 21.0±0.0 |
| Ours(C-IBP) | 21.0±0.0 | 21.0±0.0 | 21.0±0.0 |
| SA-DQN (IBP) | 21.0±0.0 | 21.0±0.0 | -20.65±0.18 |
| **Freeway** | | | |
| Ours (IBP) | 33.2±0.19 | 33.35±0.16 | 29.1±0.17 |
| Ours (C-IBP) | 34.0±0.0 | 34.0±0.0 | 25.75±0.37 |
| **RoadRunner** | | | |
| Ours (IBP) | 44495±1165 | 44445±1148 | 23820±942 |
| Ours (C-IBP) | 35240±2628 | 34160±2176 | 18315±1468 |

Table 6: Comparison of different bound calculation algorithms and their effects on the performance of specific algorithms on Atari games. Evaluated under PGD attacks of magnitude up to $\epsilon$. Ours(IBP) is RADIAL-DQN from Table 1 in for comparison, and (C-IBP), (IBP) indicates using CROWN-IBP and IBP bounds in training respectively. Only partial SA-DQN results included due to time constraints.

20. We used $\kappa = 0.9$ for all games except RoadRunner where $\kappa = 0.8$ was used. To optimize we chose the best learning rate from $\{5 \cdot 10^{-5}, 1 \cdot 10^{-4}, 2 \cdot 10^{-4}\}$ and $\kappa$ from $\{0.5, 0.8, 0.9\}$ based on performance on Pong standard and robust training respectively.

### H.2 Procgen training details

**RADIAL**-PPO models were trained from scratch, starting with 2.5M steps of standard training followed by an 22.5M steps of robust training. We experimented with finetuning the standard agent like on our Atari-results but found the results to be similar as training from scratch. We chose to report results trained from scratch since their total compute cost is lower. For robust training we used an $\epsilon$-schedule that starts as an exponential growth from $\epsilon = 10^{-10}$ and transitions smoothly into a linear schedule before plateauing at $\epsilon = 1/255$. Full details about $\epsilon$-schedule and hyperparameters can be found in our code submission. We used a constant $\kappa = 0.5$ for all models, as this default value worked well and we did not experiment with tuning it. Models were trained on a server with GeForce RTX-2080 GPU or NVIDIA Tesla P100 GPU, taking around 4 hours per standard agent and 8 hours per **RADIAL**agent in both cases. In total we estimate the compute cost for Procgen experiments(including initial tuning and testing) to be around 200-300 GPU hours.

### H.3 MuJoCo training details

For MuJoCo environments, we use a total of 4096k steps including 1024k standard steps and 3072k adversarial steps. Similar to Procgen setup, a exponential growth $\epsilon$-schedule is followed by a linear schedule for robust training. $\kappa = 0.5$ is used for MuJoCo models. Due to the compact size of MuJoCo agents, all the computation are performed on CPU. For each model with training 4096k steps, it takes around 1.5 hours on a AMD 3700X CPU. Noticeably **RADIAL**-PPO is based on faster IBP perturbation, the computational time is empirically 2/3 of the CROWN-IBP based SA-PPO method on this environment.

## I Additional results

Table 6 compares the effects of bound calculation algorithm on model performance on Atari games. We can see our algorithm performs similarly using the cheaper IBP bounds as it does on computationally expensive CROWN-IBP, whereas SA-DQN still works using IBP bounds on Pong but fails completely on the more challenging BankHeist environment.

| | Model/metric | Nominal | | PGD attack | | GWC | ACR |
|---|---|---|---|---|---|---|---|
| | $\epsilon$ | 0 | 1/255 | 3/255 | 5/255 | 1/255 | 1/255 |
| *BankHeist* **Baselines:** | | | | | | | |
| Standard | DQN [17] | 1325.5±5.7 | 29.5±2.4 | 0.0±0.0 | 0.0±0.0 | 0.0±0.0 | 0.000 |
| | A3C | 1109.0±21.4 | 1102.5±49.4 | 534.5±58.2 | 115.0±27.8 | 0.5±0.5 | 0.000 |
| Robust | RS-DQN [18] | 238.66 | 190.67 | N/A | N/A | N/A | N/A |
| | SA-DQN [17] | 1237.6±1.7 | 1237.0±2.0 | 1213.0±2.5 | 1130.0±29.1 | 1196.5±9.4 | 0.976 |
| **Our Methods:** | | | | | | | |
| | RADIAL-DQN(A#1) | 1318.5.5±4.4 | **1268.5±18.9** | **1258.0±12.5** | 1063.5±16.6 | **1232.5±35.2** | 0.814 |
| | RADIAL-DQN(A#2) | **1349.5±1.7** | **1349.5±1.7** | **1348±1.7** | **1182.5±43.3** | **1344.5±1.8** | 0.981 |
| | RADIAL-A3C(A#1) | 760.0±46.5 | 704.5±56.2 | 517.0±62.9 | 313.5±59.6 | 445.5±74.0 | 0.627 |
| | RADIAL-A3C(A#2) | 1036.5±23.4 | 975±22.2 | 949±19.5 | 712±46.4 | 851.5±2.9 | 0.718 |
| *Freeway* **Baselines:** | | | | | | | |
| Standard | DQN [17] | 33.9±0.07 | 0.0±0.0 | 0.0±0.0 | 0.0±0.0 | 0.0±0.0 | 0.000 |
| Robust | RS-DQN [18] | 32.93 | 32.53 | N/A | N/A | N/A | N/A |
| | SA-DQN [17] | 30.0±0.0 | 30.0±0.0 | 30.05±0.05 | 27.65±0.22 | 30.0±0.0 | 1.000 |
| **Our Methods:** | | | | | | | |
| | RADIAL-DQN(A#1) | 21.75 ±0.28 | 21.75 ±0.28 | 21.75 ±0.28 | 21.75 ±0.28 | 21.75 ±0.28 | 1.000 |
| | RADIAL-DQN(A#2) | 33.2±0.19 | **33.35±0.16** | **33.4±0.13** | **29.1±0.17** | **33.25±0.24** | 0.998 |
| *Pong* **Baselines:** | | | | | | | |
| Standard | DQN [17] | 21.0±0.0 | -21.0±0.0 | -21.0±0.0 | -20.85±0.08 | -21.0±0.0 | 0.000 |
| | A3C | 21.0±0.0 | 21.0±0.0 | 21.0±0.0 | -17.85±0.33 | -21.0±0.0 | 0.000 |
| Robust | RS-DQN [18] | 19.73 | 18.13 | N/A | N/A | N/A | N/A |
| | SA-DQN [17] | 21.0±0.0 | 21.0±0.0 | 21.0±0.0 | -19.75±0.1 | 21.0±0.0 | 1.000 |
| **Our Methods:** | | | | | | | |
| | RADIAL-DQN(A#1) | 9.9±3.6 | 13.7±3.0 | 13.25±3.2 | 1.1±4.5 | 2.45±4.3 | 0.950 |
| | RADIAL-DQN(A#2) | **21.0±0.0** | **21.0±0.0** | **21.0±0.0** | **21.0±0.0** | **21.0±0.0** | 0.894 |
| | RADIAL-A3C(A#1) | 20.8±0.09 | 20.9±0.07 | 20.8±0.09 | **20.8±0.09** | 20.8±0.09 | 0.982 |
| | RADIAL-A3C(A#2) | **21.0±0.0** | **21.0±0.0** | **21.0±0.0** | **21.0±0.0** | **21.0±0.0** | 0.755 |
| *RoadRunner* **Baselines:** | | | | | | | |
| Standard | DQN [17] | 43390±973 | 0.0±0.0 | 0.0±0.0 | 0.0±0.0 | 0.0±0.0 | 0.000 |
| | A3C | 34420±604 | 31040±2173 | 3025±317 | 350±93 | 0.0±0.0 | 0.000 |
| Robust | RS-DQN [18] | 12106.67 | 5753.33 | N/A | N/A | N/A | N/A |
| | SA-DQN [17] | 45870±1380 | 44300±1753 | 20170±1822 | 3350±335 | 0.0±0.0 | 0.602 |
| **Our Methods:** | | | | | | | |
| | RADIAL-DQN(A#1) | 40815±2347 | 4240±503 | 0.0±0.0 | 0.0±0.0 | 0.0±0.0 | 0.0 |
| | RADIAL-DQN(A#2) | 44495±1165 | **44445±1148** | **39560±1621** | **23820±942** | **45770±1622** | 0.994 |
| | RADIAL-A3C(A#1) | 32545±1414 | 30930±1696 | **28690±1012** | **29485±1056** | **28050±1807** | 0.932 |
| | RADIAL-A3C(A#2) | 34825±981 | 31960±933 | **29920±1496** | **31545±1480** | **31885±1912** | 0.923 |

Table 7: Full results including Approach #1. We can see Approach #1 performs well on some games but poorly on others and is outperformed by Approach #2 in all games tested. We report the mean reward of 20 runs as well as the standard deviation. We boldfaced our methods that beat or tie the best baseline. All the robust models are trained with $\epsilon = 1/255$.

In Table 7 we show the results on Atari including our Approach#1. The performance of $A\#1$ varies but is generally worse than $A\#2$. Table 8 shows our Procgen results when the same agents were evaluated with a stochastic policy.

Finally figures 5 and 6 display the effect number of training levels has on both training and evaluation performance. Training with 50 levels results in the best training performance, while unsurprisingly the largest number of training levels (200) maximizes evaluation performance. **RADIAL**-PPO standard performance on the evaluation set is competitive with original PPO, except when training with only 10 levels, which was challenging for **RADIAL**-PPO.

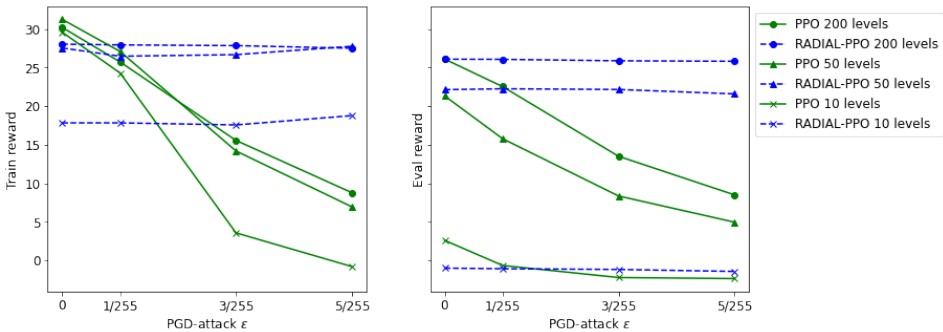

Figure 5: PPO and RADIAL-PPO performance on Fruitbot, evaluated using deterministic policy. This figure highlight the effect number of training levels has on both training and evaluation performance.

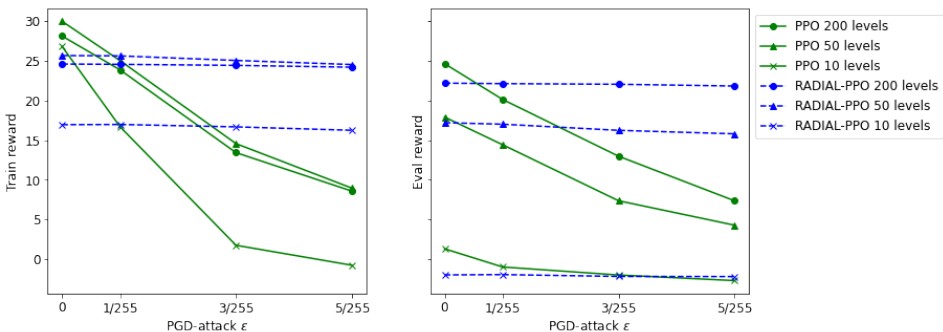

Figure 6: PPO and RADIAL-PPO performance on Fruitbot, evaluated with a stochastic policy, results similar to those observed with deterministic policy.

| Env: | Model | Distribution | Nominal | $\epsilon$=1/255 PGD | $\epsilon$=3/255 PGD | $\epsilon$=5/255 PGD |
|------|-------|--------------|---------|---------------------|---------------------|---------------------|
| Fruitbot | PPO | Train | **28.11**±**0.29** | 23.85± 0.36 | 13.43 ± 0.39 | 8.58 ± 0.35 |
|          |     | Eval | **24.63 ± 0.35** | 20.13 ± 0.40 | 12.98 ± 0.39 | 7.37 ± 0.33 |
|          | RADIAL-PPO | Train | 24.59 ± 0.31 | **24.55 ± 0.31** | **24.42 ± 0.31** | **24.19 ± 0.32** |
|          |            | Eval | 22.18 ± 0.34 | **22.12 ± 0.35** | **22.04 ± 0.34** | **21.82 ± 0.34** |
| Coinrun | PPO | Train | **9.27 ± 0.08** | **8.18 ± 0.12** | 6.71 ± 0.15 | 6.40 ± 0.15 |
|         |     | Eval | **7.99 ± 0.13** | 7.06 ± 0.14 | 6.22 ± 0.15 | 5.64 ± 0.16 |
|         | RADIAL-PPO | Train | 7.98 ± 0.13 | 7.90 ± 0.13 | **7.97 ± 0.13** | **7.94 ± 0.13** |
|         |            | Eval | 7.14 ± 0.14 | **7.21 ± 0.14** | **6.99 ± 0.15** | **6.89 ± 0.15** |
| Jumper | PPO | Train | **8.90 ± 0.10** | **8.35 ± 0.12** | 6.83 ± 0.15 | 5.43 ± 0.16 |
|        |     | Eval | **5.84 ± 0.16** | **5.86 ± 0.16** | 4.96 ± 0.16 | 4.55 ± 0.16 |
|        | RADIAL-PPO | Train | 8.09 ± 0.12 | 8.21 ± 0.12 | **8.12 ± 0.12** | **8.21 ± 0.12** |
|        |            | Eval | 5.52 ± 0.16 | 5.55 ± 0.16 | **5.61 ± 0.16** | **5.53 ± 0.16** |

Table 8: Results on the ProcGen environments with a stochastic policy. Each model was evaluated for 1000 episodes on the training/evaluation set. Reported means together with standard error of the mean. Results similar to those with deterministic policy but both agents perform better on CoinRun and Jumper and worse on Fruitbot.