# OpenReview forum: "Robust Deep Reinforcement Learning through Adversarial Loss"
_NeurIPS.cc/2021/Conference — NeurIPS 2021 Poster_

### Official Review · Reviewer_8ojS · 2021-07-14

**Rating:** 3
**Confidence:** 3

**Summary:**

The paper introduces a new robust AI approach to strengthen the RL model because tiny perturbation on input to the policy network could greatly influence the action that the target agent takes.

**Ethical Concerns:**

there are no ethical issues with this paper.

**Limitations And Societal Impact:**

The authors have adequately addressed the limitations and potential negative societal impact.

**Main Review:**

The paper states RL imposes unique challenges for using existing robustness mechanisms to the domain of RL. These challenges include credit assignments and a lack of stationary training. The deep RL algorithm learns a policy. The policy is a deep neural network. Launch an adversarial attack in the context of classification or RL has no big difference. That is to perturb the input to the neural network. To the best of my knowledge, while applying existing certified AI methods to strengthen a neural network, the certified methods should not be influenced by the training process nor the loss function but the neural network only. While reading this paper, I tried to figure out how the so-called unique challenges could impact the effectiveness of certified learning. The authors spend quite a lot of space on introducing the background of RL and some space to introduce adversarial attacks. I would suggest the authors could provide me with a more concrete analysis (maybe an example) demontrating why an existing certified learning method for a classification task does not work for the RL context. Without a clear demonstration and convincing explanation, it is hard for me to change my score.

**Time Spent Reviewing:**

2.5 hours

---

> ### Author Response · Authors · 2021-08-10
> **Author response**
>
> Thank you for the feedback! Based on your comments, we believe there might be some misunderstanding here on the focus of our method. We will try to clarify below and we welcome your further feedback for additional questions.
>
> First, we believe there may be a misunderstanding on the verification/certification v.s. certified defense. The verification and certification of DNN focuses on providing a robustness certification such that the DNN model behaves properly. In this setting, the DNN model is frozen, and the certification algorithm will output a number (known as robustness certificate) for a given DNN. There have been many works on verifying DNN models in the classification setting such as Fast-Lin[12],  Mixed Integer Programming [32] and the IBP-bounds used in [14]. These verification algorithms can be directly applied on a single step of an RL policy. However, just applying these verification algorithm is often not sufficient for creating robust systems for a few reasons:
> * These algorithms give bounds that are usually either too loose or too computationally expensive to be practically useful on a larger network.
> * Even if the bounds are tight, if the network is not robust to begin with, verifying will not make it more robust. For example, the fast IBP-bounds we use in our work cannot even verify our standard PPO network to be robust against l_{inf} bounded input perturbations with magnitude 10^(-8), six orders of magnitude smaller than our goal.
>
> On the other hand, the certified defense makes use of the certification bounds and includes them in the training for the goal of obtaining a more robust NN. In other words, the output of a certified defense algorithm is a trained DNN model unlike the verification method.
> This is done by changing the training objective(loss) of the neural network to be a function of the output bounds given by certification algorithms described above. As the bounds given by these algorithms are differentiable functions of network weights we can then minimize this loss with respect to network weights.
>
> [14] is a representative work of this method for classification, and shows that minimizing an upper bound of the original cross-entropy loss can be used to effectively train a more robust network for which the IBP-bounds are also very tight. As these methods alter the entire neural network training process, their effectiveness depends on the loss function and the training process. As such it is not trivial to apply these methods in the reinforcement learning setting.
> In a way, our Approach #1 can be seen as the most direct way to apply the idea from certified training in classification to reinforcement learning, where we replace the original loss function with its upper bound. However simply following the procedure of [14] does not work for reinforcement learning. For example, [14] starts the robust training right from the start of training. Doing this for reinforcement learning results in the network not learning a useful policy at all, and getting stuck with 0 reward for many games. Instead we found it essential to have a warmup period of standard training before we can begin robust training. We believe this is because reinforcement learning does not have a training set or correct answers, and enforcing robustness of a bad policy can make it hard to ever explore more important parts of the state-space.
>
> In addition we note that the design of the loss function is very important for the process of certified training. Even taking the upper bound of the original loss function can be complicated as it is not always clear which terms should be bounded and there are some design choices that can be made while remaining an upper bound. Second, we are not restricted to upper bounds of original loss and as we show, optimizing a different loss function (Approach #2) is often more effective. This difference between the results of Approach #1 and Approach #2 which can be seen in Table 5(Appendix) highlights the importance of the loss function design as that is the only difference between the results. In addition, the main difference between our method and SA-DQN is the design of loss function, and Table1 +Table 4(Appendix) show that we outperform SA-DQN when using the same algorithm for calculating the verified bounds.
>
> We hope this helps explain our contributions, but if you have further concerns please reach out during the discussion period.

---

> ### Author Response · Authors · 2021-08-26
> **Please let us know if you have additional questions or concerns**
>
> Dear Reviewer 8ojS, thank you again for the initial review. As the discussion period is close to the end and we have not yet heard back from you, we wanted to reach out to see if you have any additional questions or concerns. We believe there may be some misunderstandings based on the review comments, and we hope our response has addressed your concerns, but please let us know if this is not the case.
>
> We would be happy to answer any questions/concerns you may have, and we hope to use this time to clear up any remaining misunderstandings between us about our research or the contents of your review. Thanks for your time!

---

### Official Review · Reviewer_hFQ2 · 2021-07-15

**Rating:** 9
**Confidence:** 5

**Summary:**

The paper presents a well-thought and clear paper on adversarial deep RL that robustifies RL against adversarial attacks. Experimental evaluation is thorough on several benchmarks.

**Main Review:**

The presented motivation and writing is clear and concise. This work is clearly well thought through and well executed with several comprehensive experiments. The algorithm looks valid and the results are impressive. Overall, this is a solid paper with little to no flaws in my opinion (hence the short review). There are several novel contributions -- formulation of RADIAL, combination with several SOTA RL methods, new evaluation metrics, comprehensive experiments. Perhaps the only thing that could make the paper better is some real-world experiments, which is discussed in section 5.

 One minor concern is the similarity to RARL [21]. Discussion or comparison with it may further strengthen the paper.



**Time Spent Reviewing:**

1 hour

---

> ### Author Response · Authors · 2021-08-10
> **Author response**
>
> Thank you for your encouraging feedback!
>
> ***Comparison to RARL***
>
> Thank you for the reference, we will include additional discussion about our comparison with RARL[21] in the revised manuscript . The main difference between our work and [21] is the use of a different threat model. RARL focuses on defending against an adversary that can apply (bounded) forces that try to stop the agent from reaching its goals. This is motivated by the idea that robustness against this threat model will help generalization to different conditions and from simulation to reality. In contrast, our model does not experience any additional forces, but the state inputs it receives can be adversarially perturbed. This threat model aims to account for things such sensory noise. In addition, [21] train an adversary to defend against, while our method relies on certified bounds that can defend against any adversary in a family without need to train any explicit adversary. Since our goals are different, we have not compared performance against RARL in the manuscript, but we believe it would be interesting for future work to see if the two could be combined to create a more universally robust model.

---

### Official Review · Reviewer_naEG · 2021-07-16

**Rating:** 7
**Confidence:** 3

**Summary:**

This paper proposes a class of defenses for adversarial norm-ball perturbations on observations in RL, compatible with a diverse range of algorithms such as DQL, A3C, and PPO.  They also provide a new evaluation criterial, greedy worst-case reward, to evaluate their approach.

**Limitations And Societal Impact:**

Beyond the reproducibility, the main limitation I see to this work is that the attack losses are not considered sequentially, and so it is possible that these losses can compound over time.  I do not think that it's necessary to solve this for the paper to be valuable, but I think it should be highlighted as a limitation and a place for future work, so readers do not get an impression that the defense is exhaustive.

**Main Review:**

This paper integrates successful work from supervised settings into defenses for RL in a way which, to the best of my knowledge, is novel and stands to elevate work in the subfield.

The biggest concern is whether the results are replicable.  Mainly, (a) are the results statistically significant and (b) are to what extent was architectural tuning and hyper parameter search responsible for the differences.  For an example of (a), there are overlaps on some of the standard deviations of some of the results (like Fruitbot \epsilon=1/255 in Figure 2), and they are not highlighted, so it's hard to tell how many of the results have these sorts of overlaps.  Bar charts might make this easier to see, or a highlighting scheme which highlights the overlaps.  For (b) there should a discussion of what the procedure architectural tuning and hyper parameter search and why we should expect these results to be about the method itself.


minor comments:
* on line 67 there is a claim that all existing work on adversarial attacks in RL deals with norm-ball perturbations.  This is not true, as there is a line of research around adversarial attacks in the form of policies of other agents in the same environment, for instance:

Gleave, Adam, et al. "Adversarial Policies: Attacking Deep Reinforcement Learning." International Conference on Learning Representations. 2019.

Sun, Yiwei, et al. "Node injection attacks on graphs via reinforcement learning." arXiv preprint arXiv:1909.06543 (2019).

Ilahi, Inaam, et al. "Challenges and countermeasures for adversarial attacks on deep reinforcement learning." arXiv preprint arXiv:2001.09684 (2020).

* on line 77, I don't know what is meant by "these works do not study how to train more robust deep RL agents", since it seems that is exactly the point of those papers
* on line 143 it's unclear to me what is meant by "strict", and why minimizing this upper bound would imply minimizing the true function.
* I couldn't find the definition of \bar{A_Q} mentioned on line 164
* In table the best methods should be bolded regardless of if they are that of the authors

------
** Update to review **

Thank you for your detailed response to my review.  Your response addressed the bulk of my concerns in points #1 and #2.  I also appreciate the detailed response to points #3-#8, and believe the proposed changes will improve the paper.

More specifically, the detailed comparison between architectures in point #2 made it easy for me to check with past work and compare 1 to 1 that there was more tuning in past architectures, and convinced me that any difference would not explain the results.  Moreover, in point #1 the promise to fix the presentation of the statistics and to fix the bolding to make it clear to readers which results are significant and which are not should greatly improve the clarity of the results.  Given these changes, I am increasing my score.




**Time Spent Reviewing:**

4

---

> ### Author Response · Authors · 2021-08-10
> **Author response**
>
> Thank you for the feedback and raising important questions.
>
> ***#1 Reproducibility on statistical significance***
>
> Thanks for your comments! Regarding your concern regarding the statistical significance, we agree that reproducibility is very important and in fact in our manuscript, we have already reported the means of 20-1000 runs depending on the environment. Therefore, the statistical significance is already visible in all our tables (by reporting the std deviations/standard error of the mean).
>
> We agree that our current presentation may not highlight it well, and we will fix this by boldfacing all the results that are less than a standard error of the mean away from the best result. We will also change Table 1 to highlight best methods regardless of who made them as suggested.
>
> We also believe there’s some misunderstanding regarding the example you gave us on the Fruitbot $\epsilon = 1/255$ -- “For an example of (a), there are overlaps on some of the standard deviations of some of the results (like Fruitbot \epsilon=1/255 in Figure 2), and they are not highlighted” → the Fruitbot result is in Table 2 not Figure 2, and there’s no overlaps on the error margin of the Fruitbot on $\epsilon = 1/255$. For your reference, the numbers we reported are
>
> PPO Train: 25.72+-0.33, Radial PPO Train 27.93+-0.25
> PPO Eval: 22.53+-0.38, Radial PPO Eval: 26.06+-0.3.
>
> Nevertheless, we do acknowledge there are few cases overlapping within one standard error of the mean, and the detailed analysis is compiled below. As these overlaps are quite rare, they do not decrease our confidence towards RADIAL-agents outperforming baselines. We will follow your suggestion and add the above discussion in the revised manuscript.
>
> To explain our methodology, the +- numbers we have reported in Tables 1 and 2 do are standard error of the mean (abbreviated as sem below), defined as standard deviation of the measurements divided by $\sqrt{n}$, describing the uncertainty in our estimate of the mean. This is the correct way for measuring whether our observations of higher mean reward are statistically significant. In Table 3 we have reported the standard deviation instead, but for clarity and consistency we will change this to the standard error of the mean in the revised submission. Since we did 50 runs in Table 3, this will result in dividing the error measure by around 7.
>
> We have compiled a summary in Table R3 below to report on the overlap between these confidence intervals. We have defined there to be an overlap if by changing each result by less than one sem can change their order. For example, 100+-50 and 190+-50 would have an overlap, while 100+-30 and 190+-55 would not.
>
> Table R3: Summary of our results and statistical significance
>
> | Environment      | Ours better  (no overlap) | Overlap/tie | Any baseline better (no overlap) |
> |------------------|---------------------------|-------------|----------------------------------|
> | Atari(Table 1)   | 12/20                     | 7/20        | 1/20                             |
> | Procgen(Table 2) | 16/24                     | 4/24        | 4/24                             |
> | MuJoCo(Table 3)  | 5/12                      | 3/12        | 4/12                             |
>
> Discussion of Table R3: For Atari, there are 20 results we want to maximize(4 games and 5 metrics each: Nominal, 3x PGD attack and GWC). We can see that RADIAL-DQN is better with statistical significance on 12 out of 20; unclear or a tie: 7 out of 20; worse than a baseline: 1 out of 20. We think this is sufficient to indicate our method clearly outperforms previous methods on Atari games. Out of the 7 unclear results, 4 were ties on Pong where both SA-DQN and RADIAL-DQN reached the perfect result. On Procgen we observed similar results, with RADIAL-PPO only losing to PPO on standard performance, and being significantly better on 16/18 attack evaluations. Finally, we have the results from MuJoCo, which still indicate our agent performs better than baselines but are less clear as there was more variance in performance between tasks.
>
> We would like to point out that Table R3 is comparing against all baselines at once, and the best baseline is often different for different metrics. A comparison against any single baseline would be better for our algorithm than those reported on Table R3.
>
> ***#2 Reproducibility on Architecture and hyperparameter tuning***
>
> Thank you for the comment, we refer the reviewer to Appendix E, sections DQN hyper-parameters and A3C hyper-parameters(line 527-537), where we have described the procedure used for hyperparameter tuning, but will also expand on it below. In short, we only use default architectures, and tune a small set of hyperparameters based on performance in one game. We believe this is an equal or smaller amount of tuning than was done by our baselines.
>
> We tuned most hyperparameters based on what performs the best for standard training on a single game. These include parameters like learning rate, batch-size to speed up training and exploration-$\epsilon$ for DQN. These parameters were then held fixed for other games, and for robust training we only tuned the training algorithm specific parameters such as epsilon schedule and coefficients of the loss function. These were again tuned on one game and held fixed for the rest.
>
> The only exception to this is RoadRunner which was a bit unstable on default params. This is similar to hyperparameter usage in RS-DQN[18], where they had their hyperparameters fixed with the exception of small change for one game. In contrast, the baseline SA-DQN [17] found a different set of hyperparameters for each game. This leads us to believe our hyperparameter search was similar or less powerful than that used in our baselines.
>
> In terms of architecture, our RADIAL-DQN used the exact same architecture as SA-DQN to make sure the performance difference is caused by the training algorithm. For A3C we used the same architecture as the standard training repository our work was based on, except we removed the LSTM layer. For ProcGen we used the same architecture as the original paper introducing ProcGen[34]. For Mujoco results we used the same architecture and hyperparams as SA-DQN. The only difference is for Walker2D, we use 4096k steps instead of 2000k and $\epsilon_{train}=0.05$ for better comparison with other games and different attack adversaries. No architecture tuning was done by us at any point.
>
> ***#3 Threat model Line 67***
>
> We thank the reviewer’s comment and we’ll revise the sentence to “this threat model is adopted by many of the existing works investigating adversarial robustness of deep RL”. In fact, we discuss works with different threat models on lines 62-65, but we thank for the additional references and will include those into our discussion.
> Gleave et al.[S1] present the adversarial policy threat model, describing potential vulnerability of agents against adversarial models in the same environment. This is an interesting angle and a good addition to our discussion. For S2, we don't think it is very relevant for this section or our study as it is focused on using reinforcement learning to attack graphs, while our focus is on defending reinforcement learning agents.Finally, Iahi et al.[S3] provides a comprehensive overview of different attack and defense models on reinforcement learning, and we will include the reference in our discussion.
>
> [S1] Gleave, Adam, et al. "Adversarial Policies: Attacking Deep Reinforcement Learning." International Conference on Learning Representations. 2019.
>
> [S2] Sun, Yiwei, et al. "Node injection attacks on graphs via reinforcement learning." arXiv preprint arXiv:1909.06543 (2019)
>
> [S3] lahi, Inaam, et al. "Challenges and countermeasures for adversarial attacks on deep reinforcement learning." arXiv preprint arXiv:2001.09684 (2020).
>
> ***#4 Line 77, “these works do not study how to train more robust deep RL agents”***
>
> We apologize for the clarity. What we meant by line 77 is that the work [22] and [25] do not study robust RL training as we do. Instead, they are either focusing on formal verification to compute robustness certificates of a NN control policy in the feedback loop as in [22] or doing post-processing for a fixed policy to defend against adversarial perturbations as in [25]. In other words, the neural network RL policies are *fixed/frozen* in [22] and [25] (i.e the weights of NN policies are fixed), while our work is trying to *train* a robust RL policy to defend against adversarial perturbations, where the purpose of the training procedure is to find good weight parameters for the policy.
> We highlight the difference that our method aims to find a training procedure that gives a set of weights with high robustness, and is thus complementary to the approaches in [22] and [25]. We’ll clarify this in the revised manuscript.

---

> > ### Author Response · Authors · 2021-08-10
> > **Author response 2/2**
> >
> > ***#6 Definition of $\overline{A_Q}$***
> >
> > The definition is $\overline{A_Q}$ is an upper bound of the quantity $max_{||\delta|| < \epsilon} A_Q(s+\delta, a)$, calculated with IBP bounds over the network that outputs $A_Q$.
> > This was not explicitly defined due to space constraints as it follows a similar formula to other variables in the manuscript of being an upper bound of $A_Q$ defined on line 109 under $\epsilon$ perturbation. We can include an explicit definition in the revised manuscript if it helps clarify.
> >
> > ***#7 Limitation on compounding attack***
> >
> > Thanks for your comment! We agree that addressing compounding attacks is an important direction for the future and not currently addressed by our method, which will be addressed in the final paper.  We did not include compounding attacks originally for a few reasons:
> > *The compounding attack is not used by our baselines, so in order to have a fair comparison, we follow the attack methods they used.
> > *Designing strong compounding attacks is still an open problem. There are many technical and computational challenges (for example, one way to design compounding attacks is to first learn the dynamics model of the environment. However, how to learn a good dynamics model itself is still an unsolved problem). We believe there are more challenges if we want to further include it in training.
> >
> > However, following your suggestion, we have tried to conduct an experiment by using a compounding attack described in [8] against our trained models on MuJoCo, to gauge how big of a limitation this is.
> >
> > Table R4: Compounding attack experiments on MuJoCo agents trained by RADIAL-PPO
> >
> > | MuJoCo Environments | No attack    | MAD $\epsilon=0.075$ | 3-step compound , $\epsilon=0.075$ |
> > |---------------------|--------------|---------------|----------------------------|
> > | Walker2D            | 5251.6+-73.4 | 4474.7+-994.2 | 4349.9+-898.0              |
> > | Hopper              | 3737.5+-31.7 | 3252.1+-720.6 | 3439.1+-497.0              |
> > | Half_cheetah        | 4724.3+-74.7 | 4480.0+-472.9 | 4382.5 +- 1010.0           |
> >
> > We managed to use a compounding attack for our RADIAL-PPO on the continuous control task. Here is a brief description of our approach to compounding attack:
> >
> > 1. given the MuJoCo environment, learn a model $F$ for the dynamical system where $F$ approximates the next state given current state and action, $s_{i+1} \approx F(s_{i}, a_{i})$
> > 2. Given current state at time $i$ $s_i$, and a learned policy model $\pi$ from RADIAL-PPO training, forward next n compounding steps to get the final state $s_{final}$.
> > 3. We then apply adversarial attack to $s_{i}$, such that the last step is deviated from original simulation $s_{final}$ as far as possible. Craft the adversarial state perturbations $\delta s_{i}$ for the current state $s_i$, then forward next n compounding steps and solve an constrained optimization problem with PGD (e.g. Eq 3 in [8] minimizes the trajectory distance instead of final step) based on our learned $F$ in step 1.
> >
> > Table R4 shows our preliminary results for compounding attacks. Here the compounding step is chosen as 3. We found the attack strength is similar compared with the MAD attack while there are few improvement points. First, the design of the target function can be changed. Our current approach is to deviate the trajectory, a more reasonable one could be to design a loss function fooling the agent to fail. Secondly there is room for improvement by tuning our learned dynamics model and compounding attack hyper-parameters. However this initial result increases our confidence that our current training helps defend against compounding attacks. We will include above additional results and discussion in the revised manuscript.
> >
> > ***#8 Summary***
> >
> > * In #1 we addressed concerns about statistical significance, and provided a summary showing most of our results are significant
> > * In #2, we explained our process for hyperparameter tuning and (lack of) architecture tuning
> > * In #3, we clarified our comment regarding choice of threat model and discussed suggested related works
> > * In #4, we clarified distinction between [22, 25] from our work
> > * In #5, we explained line 143 and how we will adjust it
> > * In #6, we provided an explicit definition of $\overline{A_Q}$
> > * In #7, we addressed comment regarding lack of compounding attacks, and provided initial experiments showing our MuJoCo agent is still robust against a type of compounding attack

---

> ### Author Response · Authors · 2021-08-26
> **Thank you for the updated comments**
>
> We thank you for the precious feedback and we are glad to hear our response was able to address your main concerns. We will definitely revise our manuscript as discussed and we believe this will noticeably strengthen and clarify our manuscript. Thank you again!

---

### Official Review · Reviewer_Hgqp · 2021-07-19

**Rating:** 7
**Confidence:** 2

**Summary:**

The paper proposed a framework to learn RL agents that are robust to lp-norm adversarial attack. Based on existing robustness verification bounds for neural networks, the paper proposed two strategies (constructing upper bound of the loss, and minimizing the overlap between output bounds) and integrate them with three popular RL methods (DQN, A3C and PPO). The paper also propose an efficient method to approximate the worst-case reward, which is generally NP-hard to compute. Extensive experiments demonstrate the efficiency and effectiveness of the proposed methods in most of the experiments.

**Limitations And Societal Impact:**

The potential negative societal impact has been properly discussed.

**Main Review:**

The paper is an interesting application of the robustness verification bounds for neural networks to address the lp-norm adversarial attack problem in the context of reinforcement learning. I think the paper is well-structured and contains interesting discussions to this important problem. Below are some questions/detailed comments:
- At the end of Sec 2.1, the paper claims that "our framework is not limited to `p-norm perturbation and in fact can be easily extended to semantic perturbations (e.g. rotations, color/brightness changes, for vision-based deep RL agents (e.g. atari games) by leveraging the techniques proposed in [26]." I think if this can be achieved, then this may be a very important step towards robust RL against adversarial attack. From my perspective, certified defense for unrestricted (to lp-norm) adversarial attack is much more challenging if not impossible. And the bounds used for deriving the proposed method heavily relies on the lp-norm attack assumption. So please make sure this is not an overclaim and more detailed discussions on this will be appreciated.
- The motivation/rationale behind the design in Sec 3.1 - Sec 3.3 need to be elaborated in more details. Right now the paper directly presents the definitions of the adversarial loss without detailed explanation and I spent a considerable amount of time parsing the equations.

For the method in Sec 3.1 (and Sec 3.2), the logic seems to be, whenever a sampled action is better than other action, we want to make sure it is still better than that action by a margin. Here are the questions: (1) This is only for one direction. What about the other direction: whenever an action has worse Q value than another action, we want to keep this consistent and make sure it is still worse than that action? (2) Why the margin is designed to be half of Q_diff? Is the algorithm sensitive to this choice?

For the method in Sec 3.3, I found it difficult to understand the loss function. But basically we want to get a conservative estimate of the action probability, based on whether it is an optimal action. More explanations needed here.

- From Sec 3.1 to Sec 3.2, it seems Approach #2 has better performance than Approach #1. Any insights on this? Do you think this is because Approach #2 has some computational advantages? And is it possible to apply Approach #2 to PPO, e.g. by replacing the sum over actions with an integral and using sample-based approximation?

**Time Spent Reviewing:**

3

---

> ### Author Response · Authors · 2021-08-10
> **Author Response 1/2**
>
> Thank you for your valuable feedback, please see our reply below.
>
> ***#1 Beyond $l_p$-norm attack threat model***
>
> First we would like to address your concern regarding extending our work beyond $l_p$-norm bounds. To clarify, we are not claiming robustness to unrestricted attacks, but against well defined adversarial perturbations from a family different from $l_p$-norm perturbations. The basic idea of [Mohapatra et al. 2020, ref 26] is to transform a family of semantic perturbations from the pixel RGB space, and express the transformation as neural network layers (the so-called *semantify layers*, see their Eqns 5-8).  They show that many popular semantic perturbations can be expressed as semantic layers, including Hue, Saturation, Lightness, Brightness, Contrast, Rotations, Translations).
>
> Note that the commonly-used $l_p$-norm perturbation is in the *pixel RGB space*, while [Mohapatra et al. 2020] show that the semantic perturbations can be considered in the *semantic space* with $l_{\infty}$ perturbations (e.g. a change in saturation of less than 0.05). Hence, the standard $l_p$-norm verification tools can be applied to verify robustness against a family of semantic perturbation attacks, as the verification problem is restructured in a way that the change in say saturation becomes the input.
>
> Therefore, the result of [26] suggests that we can train our Radial-RL robust against semantic perturbations by simply adding semantify-layers during training, which is directly compatible with our current Radial-RL training procedure as we only need to propagate the IBP bounds starting from the semantic space (as opposed to RGB space, which is what we are doing now for the $l_p$-threat model for RGB perturbations).
>
> [26] J. Mohapatra et al., “Towards verifying robustness of neural networks against a family of semantic perturbations,” CVPR 2020
>
> ***#2 Response to question (1) about only one direction of Approach#2 loss function***
>
> Thanks for your insight! It’s true that in a sense our current  radial loss(Eqn. 5 and 6) only considers one direction. But it is worth noting that if the action taken has the highest Q-value(which happens most of the time with $\epsilon$-greedy exploration), to minimize this loss function we both want to increase the lower bound of $a$ and decrease the upper bounds of all the other actions $y$.  So the gradient updates will lower their upper bounds of bad actions even when not taken, in a sense achieving the desired effect.
>
> However, if we do take the action with the lowest Q-value by chance, the $L_{adv}$ term simply goes to zero, which may not be desirable. Inspired by your suggestion, we have conducted the following experiments to design a more *symmetric* version of the loss function Eqn. 5 to see if the new loss for RADIAL-DQN can increase our training performance.
>
> The new loss is designed as follows: we added a second term inside the summation of Eqn 5  with terms flipped to have the same regularization when Q-value for $a$ is lower than $y$, i.e. $Q_{diff}’=max(0, Q(s,y)-Q(s,a))$ and $Ovl’ = max(0, \underline{Q}(s,y,epsilon) - \overline{Q}(s,a,\epsilon) + Q_{diff}/2)$. We tested this new loss on BankHeist and RoadRunner for Atari. For Bankheist, it’s slightly worse nominal performance than our default RADIAL-DQN in Table 1, but a little better performance against 5/255 PGD. Full results are summarized in below Table R1
>
> Table R1: New experiments on symmetric loss function
>
> | Metric:        | Nominal | 1/255 PGD | 3/255 PGD | 5/255 PGD |
> |----------------|---------|-----------|-----------|-----------|
> | BankHeist:     |         |           |           |           |
> | RADIAL-DQN     | 1349.5  | 1349.5    | 1348      | 1182.5    |
> | Symmetric loss | 1325    | 1325      | 1325      | 1305      |
> | RoadRunner:    |         |           |           |           |
> | RADIAL-DQN     | 44495   | 44445     | 39560     | 23820     |
> | Symmetric loss | 34905   | 35315     | 33315     | 21385     |
>
> Discussion of Table R1: we hypothesize that the slightly worse performance could be caused by more strict enforcement of the gap between outputs in this new loss, which could potentially make the policy harder to correct if wrong Q-values are reached at some point. We will include the above result and discussion in the revised manuscript, thanks for your insight!
>
>
> ***#3 Response to question (2) on the choice of margin i.e. $0.5*Q_{diff}$***
>
> The main idea to choose the margin is as follows: the margin should be $c*Q_{diff}$, with $0<c<1$. If Margin=$Q_{diff}$, it is equal to the situation where the bounds are exactly tight (for example if $\epsilon=0$), and thus it's impossible to have a margin bigger than $Q_{diff}$. We did notice that it is important to require a margin > 0 as we have seen experiments with margin=0 that did not reach desired robustness. We used $c=0.5$ for simplicity, and did not tune as it worked well in our experiments. Note that SA-DQN[17] uses a constant margin requirement in a loss function somewhat similar to ours, this has undesirable effects when $Q_{diff}$ is smaller than this margin, see Appendix D for more discussion on this.
>
> Following your suggestion, we have conducted additional experiments on the two games Roadrunner and BankHeist for better understanding the sensitivity of our algorithm to choice of $c$. We observed that the choice of $c$ does affect the robust agent’s performance, significantly in some games such as RoadRunner. Full results in the table below:
>
> Table R2: Experiment on changing margin coefficient for RADIAL-DQN on Atari
>
> | Metric          | Nominal | 1/255 PGD | 3/255 PGD | 5/255 PGD |
> |-----------------|---------|-----------|-----------|-----------|
> | BankHeist:      |         |           |           |           |
> | c=0.25          | 1347.5  | 1214.5    | 1257      | 1113.5    |
> | c=0.5 (default) | 1349.5  | 1349.5    | 1348      | 1182.5    |
> | c=0.75          | 896     | 896       | 896       | 896       |
> | RoadRunner:     |         |           |           |           |
> | c=0.25          | 20385   | 15135     | 10795     | 11220     |
> | c=0.5 (default) | 44495   | 44445     | 39560     | 23820     |
> | c=0.75          | 4870    | 4870      | 4870      | 4870      |
>
> We can see that $c=0.75$ produces poor standard performance and we think it’s because this requirement is too strict and the policy collapses to some simple policy with lower reward. For $c=0.25$, the resulting robust policy works well and is only slightly worse than $c=0.5$ on both standard performance and robustness in the BankHeist; however, for RoadRunner, the results are much worse than $c = 0.5$. We will add the above study on different c to the revised paper, thanks for your suggestion.
>
> ***#4 Rationale behind sec 3.1-sec 3.3***
>
> The intuition for Approach #1, used in section 3.3 is to derive an “upper” bound of the RHS of Eqn 3 which will always hold. I.e. for all perturbation magnitude $\epsilon$, our derived $L_{adv}$ will *always* be greater or equal to the RHS of Eqn 3. In Section 3.3, the loss Eqn 7 is an upper bound of Eqn 3. In general, minimizing an upper bound of the true target is a common and effective strategy in machine learning, and is employed in for example in Latent Dirichlet Allocation [Blei et al., 2003] to optimize the models as well as training certifiably robust models in a classification setting [11, 14], which is done by minimizing the upper bound of the cross-entropy loss under $\epsilon$-bounded perturbation.
>
> However, applying Approach #1 didn’t always work as well as desired (see detailed discussion on our reply #6 Our Approach #2 v.s. Approach #1), and Approach #2 was designed as an alternative. The key of idea of Approach #2 was to only restrict what is most important to robust performance. We identified this to be the “Overlap” pictured in Figure 2, the difference between lower bound of action taken and upper bound of other actions. But not all overlap is equally important. If two actions have a very similar Q-value, we don’t really care if there is overlap between them, as taking a different but equally good action under perturbation is not a problem. To tackle this we added weighting by $Q_{diff}$, which helps by multiplying overlaps with similar Q-values by a small number and multiplying overlaps with large Q-value difference by a large number.
>
> ***#5 The loss function in section 3.3***
>
> The loss function in section 3.3 is for Radial-PPO, and the idea is to use the interval bounds to get an upper bound of the loss function. $L_{adv}$ in Eqn 7 is an upper bound of Eqn. 3 for standard PPO loss when the input $s_t$ to the current policy $\pi(a_t | s_t, \epsilon;\theta)$ is perturbed, as the only term affected by this perturbation is $\pi(a_t | s_t, \epsilon;\theta)$. Note that input to old policy is unperturbed as this is only used to help training, and we are not aiming to defend against train time adversaries. We use $\hat{\pi}$ as the perturbed (worst-case) policy. Eqn 8 describes this worst-case policy, which is designed to maximize the loss $L_{adv}$ (i.e. the goal of an adversary). For example, for positive $A_t$, lowering the output of $\pi$ will increase loss (because there’s negative sign in front of the min operator in Eqn 7); while for negative $A_t$, increasing the output of $\pi$ will increase loss. Hence, the worst-case policy is designed as Eqn 8 to maximize the adversarial loss. Minimizing this upper bound of the loss(Eqn 7) will effectively push the lower bound of action probability high when the action is good ($A_t >= 0$) and the upper bound of action probability to be low when the action is bad($A_t < 0$), which is desirable for a robust algorithm.

---

> > ### Author Response · Authors · 2021-08-10
> > **Author Response 2/2**
> >
> > ***#6 Our Approach #2 v.s. Approach #1***
> >
> > Our intuition is that, Approach #2 is designed to better focus on exactly what is important for robust performance e.g. don't change from the best action, while Approach #1 has stronger requirements that are not necessary. For example Approach #1 optimizes for tight upper bounds for the highest action and tight lower bounds for poor actions. In our experiment, we observed that the increased flexibility of Approach #2 translates to improved performance, both standard and robust.
> >
> > ***#7 Apply Approach #2 in continuous environments***
> >
> > Thanks for your suggestion! We think it’s possible to apply Approach #2 on PPO for continuous control action as you described -- by approximating the integral via uniform samples from the action space or fixed steps. We think this is a good idea and has the potential to work, and we will aim to include a study of Approach #2 in Radial-PPO in the revised manuscript and add corresponding discussion.
> >
> >
> > ***#8 Summary***
> >
> > We briefly summarized our above response to your comments below:
> > * In #1, we explained in detail how our method could be applied beyond the $l_p$ norm threat model.
> > * In #2, we performed new experiments with a more symmetric version of our Approach#2 loss function for RADIAL-DQN as suggested. We found them to have good robustness but slightly reduced standard performance compared to our original.
> > * In #3, we tested the sensitivity of our Approach #2 to the choice of margin with RADIAL-DQN on ATARI, we found that current $0.5*Q_{diff}$ performs clearly better than using a coefficient of 0.25 or 0.75
> > * In #4, we clarified our motivation and process behind our design of loss functions.
> > * In #5, we explained and clarified section 3.3
> > * In #6, we explained our intuition behind why Approach #2 performs better
> > * In #7, we addressed the good suggestion of how to apply Approach #2 on continuous control

---

### Decision · Program_Chairs · 2021-09-27

**Decision:**

Accept (Poster)

**Comment:**

I thank the authors for their submission and active participation in the discussions. The paper tackles an important problem as robustifying deep RL agents is crucial for making progress towards real-world applications. Reviewers appreciate its significance [Hgqp,naEG,hFQ2], novelty [naEG], and interesting discussion [Hgqp], as well as its thorough experiments [hFQ2]. During rebuttal and discussion, reviewer naEG was compelled by the author response and clarifications, as well as reviewer hFQ2's arguments. I am discounting 8ojS's clear stance against the paper as I believe the author response sufficiently addresses their concerns and they haven't indicate otherwise during the discussion. I also appreciate the author's openness about potential limitations of their work. Overall, I recommend acceptance and encourage the authors to further improve their paper based on the reviewer feedback.